# High-resolution mapping of circum-Antarctic landfast sea ice distribution, 2000-2018

Alexander D. Fraser[1,2,3], Robert A. Massom[4,2], Kay I. Ohshima[3], Sascha Willmes[5], Peter J. Kappes[6], Jessica Cartwright[7,8,2], and Rick Porter-Smith[2]

[1]Institute for Marine and Antarctic Studies, Hobart, Tasmania, Australia
[2]Antarctic Climate & Ecosystems Cooperative Research Centre, Hobart, Tasmania, Australia
[3]Institute of Low Temperature Science, Hokkaido University, Sapporo, Hokkaido, Japan
[4]Australian Antarctic Division, Kingston, Tasmania, Australia
[5]University of Trier, Germany
[6]Oregon Cooperative Fish and Wildlife Research Unit, Department of Fisheries and Wildlife, Oregon State University, USA
[7]National Oceanography Centre, Southampton, UK
[8]University of Southampton, Southampton, UK

**Correspondence:** Alexander Fraser (Alexander.Fraser@utas.edu.au)

**Abstract.** Landfast sea ice (fast ice) is an important component of the Antarctic nearshore marine environment, where it strongly modulates ice sheet-ocean-atmosphere interactions and biological and biogeochemical processes, forms a key habitat, and affects logistical operations. Given the wide-ranging importance of Antarctic fast ice and its sensitivity to climate change, improved knowledge of its change and variability in its distribution is a high priority. Antarctic fast-ice mapping to date has been limited to regional studies and a time series covering East Antarctica from 2000 to 2008. Here, we present the first continuous, high spatio-temporal resolution (1 km, 15 day) time series of circum-Antarctic fast ice extent; this covers the period March 2000 to March 2018, with future updates planned. This dataset was derived by compositing cloud-free satellite visible and thermal infrared imagery using an existing methodology, modified to enhance automation and reduce subjectivity in defining the fast ice edge. This new dataset (Fraser et al., 2020) has wide applicability, and is available at http://dx.doi.org/doi:10.26179/5d267d1ceb60c. The new algorithm presented here will enable continuous large-scale fast ice mapping and monitoring into the future.

## 1 Introduction

Landfast sea ice (fast ice) is a pre-eminent feature of the Antarctic near-coastal environment, where it forms a relatively narrow (several tens to ~200 km wide) zone of consolidated ice attached to grounded icebergs, coastal margins (including sheltered embayments), floating glacier tongues and ice shelf fronts (World Meteorological Organization, 1970). Depending on location, it can be either annual (forming each austral autumn-winter and melting back each spring-summer) or perennial (Fraser et al.,

2012), with multi-year fast ice attaining thicknesses up to several tens of metres (e.g., Massom et al., 2010). By forming a recurrent, persistent and highly-consolidated substrate of sea ice and snow, fast ice strongly modulates important physical and

biological processes occurring at the Antarctic coastal margin - including stabilisation of ice shelves that moderate ice sheet mass loss to the ocean and resultant sea level rise (Massom et al., 2018). Given these factors, there is strong motivation for improved knowledge of its circum-Antarctic distribution, change and variability. Indeed, the lack of a long-term and continent-wide Antarctic fast ice dataset from which to accurately gauge change and variability has been highlighted as a major gap by the Intergovernmental Panel on Climate Change (IPCC) Fifth Assessment Report (Vaughan et al., 2013) and the Special

Report on the Ocean and Cryosphere (Meredith et al., 2019).

The consistent large-scale and long-term monitoring of Antarctic fast ice from space necessitates overcoming a number of inherent challenges relating to detection and resolution (both spatial and temporal), given the attributes of the satellite data themselves, and the nature of fast ice itself. For one thing, fast ice is a narrow remote-sensing target compared to the more extensive moving pack-ice zone (that is regularly monitored by coarse-resolution satellite passive-microwave sensors), and ad-

vection of pack ice against adjacent fast ice can lead to a relatively indistinct boundary between the two. Table 1 summarises the current status of Antarctic fast ice detection and mapping from space, and the advantages and disadvantages of the techniques used (see also Lubin and Massom, 2006). Wide-swath moderate-resolution satellite visible and thermal infrared (TIR) imagery offers excellent geographical coverage at kilometre-scale resolution and on daily time-scales, but it is strongly affected by persistent cloud cover year-round and polar darkness,the latter precluding use of visible imagery in winter (Fraser et al., 2009).

While this limitation can theoretically be circumvented by using high-resolution Synthetic Aperture Radar (SAR) imagery (Giles et al., 2008; Li et al., 2018; Kim et al., 2020), the application of SAR to large-scale fast-ice mapping and time-series analysis has to date been limited in space and time by its relatively-narrow swath coverage and uneven image acquisition around coastal Antarctica. Satellite passive-microwave data, on the other hand, offer complete circumpolar coverage on a daily basis (largely unaffected by clouds and darkness), but a poorer spatial resolution of ∼6.25 km (Nihashi and Ohshima, 2015)

limits its capability for accurate fine-scale mapping of fast ice.

As a result of these challenges and factors relating to scientific focus, the mapping of Antarctic fast ice from space has to date been largely confined to limited geographical regions (e.g., around Antarctic bases and penguin colonies) and also relatively short time series or snapshots. These are based on manual interpretation of ad hoc digitizations of satellite SAR and cloud-free visible/TIR imagery (e.g., Mae et al., 1987; Ushio, 2006; Giles et al., 2008; Massom et al., 2009; Aoki, 2017; Kim et al.,

2018; Li et al., 2018; Labrousse et al., 2019; Kim et al., 2020). A significant advance in continuous coverage was made by Fraser et al. (2012) in their analysis of fast ice across East Antarctica (10° W to 172° E) based on compositing of cloud-free imagery from the MODerate-resolution Imaging Spectro-radiometer (MODIS) sensors onboard NASA's Aqua and Terra satellites, for the period 2000-2008. This study also used a more rigorous definition of fast ice that included a temporal criterion e.g., that sea ice must remain stationary for 20 days to be classified as fast ice (Fraser et al., 2010), but still involved a signif-

icant amount of time-consuming and intensive manual analysis. Considerable progress has since been made in the automated extraction of the fast-ice edge in both MODIS (Fraser et al., 2019) and SAR image products (e.g., Kim et al., 2020; Li et al., 2018), in parallel with advancements in SAR-based fast ice detection in the Arctic e.g., Mahoney et al. (2007), Meyer et al.

(2011) and Dammann et al. (2019). Improved automation is particularly important given the volume of data involved and the considerable effort that is required to manually digitize the fast ice edge using non-automated techniques (Fraser et al., 2012).

To date, large-scale and long time-series mapping of Antarctic fast ice has been confined to two datasets. These are: 1) the manually-classified MODIS-based dataset (Fraser et al., 2012); and 2) a fully automated Advanced Microwave Scanning Radiometer for EOS (AMSR-E)-derived time series for the time period 2003 to 2012 from Nihashi and Ohshima (2015). While the latter dataset is circumpolar in its coverage, an analysis by Fraser et al. (2019) shows a tendency of passive-microwave radiometry to underestimate fast-ice extent due to an inherent insensitivity to young fast ice <90 days old, and its relatively

poor spatial resolution.

    Here, we introduce and provide details of a new algorithm and dataset - the first high spatio-temporal resolution (1 km; 15 day) long-term time series (currently 2000 to 2018 with regular updates planned) of complete circum-Antarctic fast ice extent. This new technique is based on the compositing of MODIS cloud-free visible and TIR images using a technique described by Fraser et al. (2009), but improved and with automated extraction (as far as possible) of the fast ice edge through addition

of edge-detection logic. This reduces the amount of manual interpretation required while increasing the level of objectivity in retrieving the fast ice maps.

    In the next sections, we present a description of the datasets and updated methods used to transform MODIS imagery into consistent fast ice maps. Following this, we present the fast ice dataset and provide a comparison with the earlier East Antarctic fast ice time series from Fraser et al. (2012). Analysis of the time series, anomalies and trends for the entire circumpolar record

is beyond the scope of this paper, and is the subject of a study in preparation. A major aim here is to make this dataset available to the wider scientific community, thereby facilitating collaborative fast ice-related research across disciplines.

**Table 1.** Table detailing techniques used to detect and/or map Antarctic fast ice, encompassing both large-scale and case studies. MODIS: Moderate Resolution Imaging Spectroradiometer (NASA). AVHRR: Advanced Very High Resolution Radiometer (NOAA). SAR: Synthetic Aperture Radar. SSM/I: Special Sensor Microwave/Imager (Defence Meteorological Satellite Program). AMSR-E: Advanced Microwave Scanning Radiometer for Earth Observation System (Japan Aerospace Exploration Agency). AMSR-2: Advanced Microwave Scanning Radiometer-2 (Japan Aerospace Exploration Agency). ALOS: Advanced Land Observing Satellite (Japan Aerospace Exploration Agency). PALSAR: Phased Array type L-band Synthetic Aperture Radar (onboard ALOS). Envisat: Environment Satellite (European Space Agency). ASAR: Advanced Synthetic Aperture Radar (onboard Envisat). RADARSAT: Radar Satellite (Canadian Space Agency).

| Product | Large scale dataset or case study? | Instrument | Technique | Timespan | Temporal resolution | Spatial resolution | Spatial coverage | Advantages | Disadvantages | Publications |
|---|---|---|---|---|---|---|---|---|---|---|
| This work | Large scale dataset; ongoing | MODIS visible/ thermal IR | Semi-automated, composite-based | Mar 2000 – Mar 2018; updates planned | 15 day | 1 km | Circum-Antarctic | High spatio-temporal resolution; close agreement with Fraser et al. (2012); long and continuous time series; semi-automated | A degree of subjectivity remains; considerable manual overhead | This work |
| MODIS manually-digitised dataset | Large scale dataset; discontinued | MODIS visible/ thermal IR | Fully manual, composite-based | Mar 2000 – Dec 2008 | 20 day | 2 km | East Antarctica | Medium spatio-temporal resolution; continuous time series | Fully manual | Fraser et al. (2009, 2010, 2012) |
| Ad-hoc MODIS and AVHRR digitisations | Case studies | MODIS and AVHRR visible/ thermal IR | Fully manual, snapshots | Nov 1978 – present | Snapshots | 1 to 4 km | Focus regions | Long time series | Poor georeferencing and resolution at times (AVHRR), cloud-affected, fully manual, snapshots | Mae et al. (1987); Massom (2003); Ushio (2006); Massom et al. (2009); Aoki (2017); Kim et al. (2018); Labrousse et al. (2019) |
| National Ice Center charts | Large scale dataset; ongoing | Various sources (SAR, visible, TIR, scatterometer) | Fully manual, snapshots | Jul 1998 – present | Snapshots within one week | Data- and year-dependent | Circum-Antarctic | Long time series; near-real time | Unvalidated; fast ice retrieval of variable accuracy; not a consistent circumpolar product; format changes; many analysts; fully manual; coarse spatial resolution at times | N/A |
| Passive microwave spectral | Large scale dataset; ongoing | SSM/I, AMSR-E, AMSR-2 | Principal component analysis | Dec 1990 – present | 90 day | 6.25 to 12.5 km | Circum-Antarctic | Fully automated | Insensitive to young (<90 day old) fast ice; coarse spatial resolution | Tamura et al. (2007, 2016); Nihashi and Ohshima (2015) |
| Object-based SAR | Case study | ALOS PALSAR | Object-based definition | Snapshots in 2007 and 2010 | 5 day | 100 m | Various west Antarctic sites | High spatial resolution; reasonable accuracy; potential for extensive automation | Limited time series of underlying data | Kim et al. (2020) |
| SAR gradient difference | Case study | Envisat ASAR, Sentinel-1 | Automated ice edge from gradient difference | Snapshots in 2008 and 2016 | 13 to 20 day | 40 m | Prydz Bay | High spatial resolution; high accuracy; potential for extensive automation | Requires reference fast ice climatology to remove spurious edges; limited time series of underlying data | Li et al. (2018) |
| Motion-based SAR | Case Study | RADARSAT | Maximum cross-correlation | Snapshots in 1997 and 1999 | 1 to 20 day | 100 m | Western Pacific Ocean sector | High spatial resolution; potential for extensive automation | Limited time series of underlying data | Giles et al. (2008) |
| Multi-sensor fusion | Large scale dataset; discontinued | MODIS, AMSR-E, SSM/I | Machine learning | 2003 to 2008 | 20 day | 25 km | Circum-Antarctic | Novel technique; automated | Low spatial resolution; apparent fast ice overestimate | Kim et al. (2015) |

## 2 Dataset and methods

The fast ice time series presented here for the entire Antarctic coastline uses imagery from the MODIS sensors on both the Terra (MOD) and Aqua (MYD) satellites, and obtained from NASA's Level-1 Atmosphere Archive & Distribution System Distributed Active Archive Center (https://ladsweb.modaps.eosdis.nasa.gov). The first ∼2 years of this dataset was produced using only Terra MODIS imagery, prior to the July 2002 commissioning of Aqua MODIS. Specifically, the algorithm uses data from the following:

– Channel 1 (visible, 620 to 670 nm) from the MOD/MYD02QKM dataset, the 250 m resolution level 1B product being available during times of solar illumination;

– Channel 31 (thermal infrared, 10.78 to 11.28 $\mu$m) from MOD/MYD021KM, the 1 km resolution level 1B product being available regardless of sunlight and providing information during periods of polar darkness;

– The high resolution georeferencing arrays from the MOD/MYD03 product; and

– The level 2 cloud mask product (MOD/MYD35_L2).

A crucial feature of the new algorithm and time series is accurate masking of the Antarctic continent, ice shelves and nearshore islands. For this, we use the MODIS-based Mosaic Of Antarctica (MOA) coastline digitisation – both the 2003-04 product (Haran et al., 2005; Scambos et al., 2007) and the 2008-09 product (Haran et al., 2014). Change in ice-shelf front location over time due to ice-sheet advance or iceberg calving necessitates progressive updates to the MOA coastline product. For this, we make annual modifications to the location of the ice-sheet margin by manually digitising the change in the position of the ice shelf front once per year, at the time of annual climatological minimum fast ice extent i.e., day of year 061-075 (Fraser et al., 2012). Temporal compositing is required to create cloud-free images of the entire Antarctic coastal zone. The MOA-derived annual coastline rasters are also manually edited to correct an artefact in the coastline in the Vestfold Hills region, near Davis Station (68.5° S, 78.25° E). Although this process is entirely manual, it occurs only once per year so is not particularly laborious. It is possible that some very persistent multi-year fast ice is misclassified as ice shelf in limited regions, although particular care was paid to avoid this.

All swath-to-grid projection of the level 1 and 2 imagery is performed with the MODIS Swath-To-Grid Toolkit (MS2GT, version 0.26), available at https://nsidc.org/data/modis/ms2gt. We grid all level 1 and 2 products to a 1 km resolution polar stereographic grid with a latitude of true scale set to 70° S (grid size: 5625 * 4700 pixels, covering the expected maximum circumpolar fast ice extent), to maximise compatibility with other sea ice datasets from the NSIDC. We choose a 1 km spatial resolution to match the nominal resolution of the MODIS TIR channels.

We broadly follow the fast ice mapping methodology developed by Fraser et al. (2009, 2010), but with significant improvements to enhance automation and objectivity in delineation of the fast ice edge. The earlier East Antarctic work first constructed cloud-free composite images of the surface over consecutive 20 day periods, based on MODIS visible and TIR imagery and the NASA MODIS cloud mask product (Fraser et al., 2009). These composites (i.e., a TIR composite at all times of the year, and a

visible composite when solar illumination was present) were then used for manual delineation of the fast ice edge (Fraser et al.,
2010). The authors noted regions and times of lower composite image quality when persistent cloud obscured the surface in
the majority of component images (cloud is a major issue for optical remote sensing of the surface in polar regions). In the
Fraser et al. (2009) algorithm, even an optically-thin layer of clouds in which the surface features were still discernible was
excluded from the cloud-free composite image, sometimes resulting in "data holes" in the image time series. Here, we mitigate
this shortcoming by: 1) more intelligently ranking cloud content and ensuring a more uniform distribution around the Antarc-
tic coast, thereby increasing the chance of a cloud-free view of the surface; and 2) implementing automated determination
of the fast ice edge location in an independent image processing pathway which does not rely on the cloud mask product.
Here we rank all cloud-mask granules by their cloud content, and choose the 100 least cloudy granules in each of six regions
(each approximately 60 degrees of longitude wide) around the Antarctic coast for compositing and further processing, i.e.,
600 MOD/MYD02 granules in total per 15 day window. This regional consideration was implemented in an effort to ensure a
more even distribution of MOD02 granules. We found that without this consideration, the ranking algorithm resulted in a high
concentration of granules in a limited number of cloud-free regions at the expense of cloudy regions.

In the latter processing pathway described above, we perform edge detection on all individual gridded MOD/MYD02 gran-
ules, exploiting the difference in both albedo and infrared brightness temperature between ice, cloud and ocean. This is based
on the fact that both cloud and pack ice edges are dynamic between images whereas fast ice edges are likely to be relatively per-
sistent in location (i.e., stationary). We use the Canny (1986) edge detection method to ensure that edges are correctly localised
and detected only once. We then sum all edges within a 15 day window, thereby determining which edges are most persistent.
These persistent edges are then interpreted to be either the fast ice edge or the continental margin. Since the location of the
continental margin is well-known, we exclude these edges from consideration and are thus left with a representation of the fast
ice edge. This map of persistent edges over each 15 day window forms the basis for subsequent automated circum-Antarctic
fast ice edge detection.

The 15 day time-step is chosen by balancing a desire for finer resolution against the potential for pack ice temporarily ad-
vected against the coast to be misclassified as fast ice despite no mechanical fastening taking place. Around most of coastal
Antarctica, the climatological near-surface wind direction is generally offshoreward to westward (Turner and Pendlebury,
2004), thus promoting advection of pack ice away from the coast. Blocking anticyclonic pressure systems do occur in southern
mid-latitudes and these can result in persistent onshoreward winds in particular regions of the Antarctic coast, although the
residence time for such systems is rarely longer than one week (Massom et al., 2004). As such, a time-step of 15 days is suffi-
ciently long to preclude most of these cases. Drifting sea ice pinned between grounded icebergs may also be misclassified as
fast ice, though our earlier work showed that the persistent advection of pack ice into pre-existing coastal features is likely to
be a larger problem, and that pack ice held fast between grounded icebergs may quickly become fastened (Fraser et al., 2010).
Cloud coverage, which can be persistent in some regions, is a further barrier to a finer time-step when producing visible and
TIR composite images of the surface (Fraser et al., 2009).

Our image processing pipeline is outlined below, and is depicted by flow-chart in Figure 1. For each 15 day window in the
March 2000 to March 2018 study period, we:

1. Download and grid all MOD/MYD35_L2 (cloud mask) granules covering the Antarctic coastal zone (approximately 1,800 granules per 15 day interval). Outcome: A complete library of gridded cloud mask granules.

2. Rank granules by cloud content. Outcome: Ranked list of least-cloudy scenes.

3. Select the top 600 cloud-free granules, cognizant of granule location (to ensure sufficient coverage in all coastal regions). Outcome: List of 600 least-cloudy scenes with relatively even coverage around the continent.

4. Download and grid all corresponding MOD/MYD02QKM (reflectance; available during periods of solar illumination), MOD/MYD021KM (TIR brightness temperature; available year-round), and MOD/MYD03 (high-resolution geolocation data) granules. Outcome: Library of least-cloudy reflectance and TIR brightness temperature scenes, gridded.

5. Process gridded MOD/MYD02 images for manual and automated edge-detection purposes:

   – Produce cloud-free composite images from 600 input granules: Construct thermal infrared and (when solar illumination available) visible cloud-free composite images from the gridded MOD/MYD02 and MOD/MYD35_L2 granules, following Fraser et al. (2009, 2010). Outcome: Composite images.

   – Produce Canny edge images for each granule: Canny edge-detect MOD/MYD02 granules and sum within the current 15 day period. Outcome: Canny edge sum image for automated edge extraction.

   – Produce Sobel edge images for each granule (Sobel, 2014): Sobel edge-detect MOD/MYD02 granules and sum within the current 15 day period. Outcome: Sobel edge sum image to guide manual fast ice edge interpretation.

   – Produce gradient-median-composite images: Median-filter (using a 7*7 pixel sliding window) each composite image (i.e., visible and TIR), then take the absolute value of the gradient of this image, indicating edges in the composite image. Outcome: Gradient-mean-composite images for automated edge extraction.

   – Produce modified lead-detection images after Willmes and Heinemann (2015), but with a larger filtering window of 251 pixels (originally 51 pixels) to enhance contrast in regions of fast ice. Outcome: lead-detection images to guide manual fast ice edge interpretation.

6. Construct an automated classification base image:

   – Compute the per-pixel product of the Canny edge image and the gradient-median-composite image described above, which was found to accurately and correctly locate many fast ice edges (i.e., this is an original algorithm). This product represents a continuous measure of fast ice edge confidence. Outcome: base image for automated fast ice edge extraction.

   – Produce a normalised histogram of edge confidence, setting four adaptive thresholds at 0.995 (highest-confidence edge), 0.990, 0.985 and 0.980 (lowest-confidence edge). These thresholds are used to construct a grey-scale representation of the edge confidence for each pixel on the grid. Outcome: Confidence-classified automated fast ice edge map.

  – Mask the edge confidence map using the rasterised MOA coastline and write out as the automated classification base image. Multiple spurious edges exist at this point. Outcome: Coast-masked automated edge image.

7. Carry out necessary manual processing (relatively labour-intensive, one year takes approximately 40 hours):

  – Close inspection and completion of edges in automated classification base image, guided by: a) the Sobel edge image; b) cloud-free composites; and c) modified lead-detection images. This is used to: i) verify automated fast ice edge extraction, and ii) manually complete/add edges where automated extraction fails to detect the fast ice edge. Sobel edge detection is used in this manual step, rather than Canny edge detection, because it produces a broader (i.e., several pixels wide) edge which is tolerant of small changes in ice edge location. Outcome: Image of completed fast ice edges.

  – "Bucket fill" those pixels between the continental margin and the now-continuous ice edge to represent fast ice coverage (extent). Outcome: Near-final image of fast ice edge and "filled" pixels.

8. Automatically remove spurious edges (i.e., edges not adjacent to fast ice) remaining from the classified image. Outcome: Final classified fast ice image.

Since the "bucket fill" step requires a continuous fast ice edge, and because the automatically-determined fast ice edge is often incomplete, manual intervention is frequently required both to form a continuous fast ice edge and to validate the position of the automatically-determined fast ice edge. An example classification showing both manual and automated ice edge detection is shown in Figure 2. This manual intervention is relatively time-consuming and reduces objectivity to some extent, but is considered to be a fundamental step in visible/TIR fast ice extent retrieval. It should be reiterated here that the inclusion of automatic edge determination is a considerable advance from the original fully-manual final step of edge extraction described by Fraser et al. (2010). In order to mitigate the possibility of manual edge definition contributing to false trends in the dataset and following Fraser et al. (2012), all edge verification and manual edge completion is performed in a random order.

When manual edge delineation is not possible in any given region for a particular 15 day period (e.g., due to persistent thick cloud), the method employs a subjective definition of the location of the fast ice edge based on imagery from the immediately previous and/or subsequent 15 day periods, following Fraser et al. (2010). An extreme example relates to the fast ice map from DOY 166-180 in 2001, during most of which the Terra MODIS instrument was in "safe mode" and acquired no data. Here, in the interest of providing a temporally-contiguous dataset, we opt to use the fast ice map from the following timestep (DOY 181-195, 2001) but mark all edges as "manually-determined" to indicate higher uncertainty in the fast ice edge retrieval for DOY 181-195 (2001).

Determination of uncertainty for this dataset (in both edge location and resulting fast ice areal extent) requires careful consideration. The primary uncertainty arises from digitisation error, typically given in pixels, in areas of manual ice edge determination, which then propagates to an areal uncertainty value. However, neither the digitisation error nor the propagation to an areal uncertainty are straightforward to determine/quantify. Prior work made broad estimates of the manual digitisation error by carrying out an independent re-digitisation of a subset of the fast ice edge and resolving differences in the resulting

fast ice area (Fraser et al., 2010). This approach, however, requires both extrapolation of errors from a small subset to the entire dataset and duplication of time-consuming manual edge extraction. In our modified approach presented here, we employ a novel alternative approach for uncertainty estimation which addresses these shortcomings. This involves analysis of the per-pixel difference in ice edge location in two consecutive fast ice maps, for all pairs of consecutive images in the entire dataset. In the case of an automatically-extracted fast-ice edge pixel, this difference purely reflects the change in location of the ice edge (plus or minus a small, sub-pixel scale digitisation error, which we also quantify). In the case of a manually-extracted ice edge pixel, it reflects the sum of the ice edge change plus the digitisation error. Thus, to estimate the digitisation uncertainty, we:

1. assume that automatically-determined edges are accurate in location (an appropriate assumption due to excellent edge localisation of the Canny edge detection filter underpinning the automation);

2. quantify the mean fast ice edge separation between subsequent images only for automatically-determined edge pixels. We find the nearest edge of similar type. In this step, we match automatically-determined edge pixels with the nearest automatically-determined edge in the subsequent image. Cross-type edge matches are ignored (i.e., auto to manual, or manual to auto) to avoid confounding results. A cutoff of +/- 50 px (i.e., an ∼100 km window) is used as an extremely conservative upper bound to avoid the rare case of pixels matching with distant pixels. We thereby produce a mean measure of ice edge location change between two consecutive 15 day time periods;

3. as above but for manually-determined edge pixels, to produce a mean measure of ice edge change plus digitisation error; and

4. subtract the former from the latter, resulting in a digitisation error estimate for manually-determined ice edge pixels.

We also estimate the sub-pixel error in digitisation, i.e., grid-scale effects in the digitisation error. This estimation is achieved by performing 10,000 simulations of a one-dimensional random edge position and compare it to the centre location of a sample pixel. The RMS of the residual between the genuine pixel centre and the simulated centre is taken to be the sub-pixel error. Thus, the automatically-determined edge error is taken to be the sub-pixel error only, and the manually-determined edge error is taken to be the quadrature sum of the sub-pixel and manual digitisation errors. Following estimation of both the manual and sub-pixel digitisation errors, we estimate areal uncertainty for each fast ice map by:

1. ensuring that all fast ice edges are one pixel wide by performing a morphological skeleton operation;

2. weighting all skeletonised edge pixels by their respective area; then

3. multiplying by the appropriate error, as estimated above.

This approach to areal uncertainty calculation is highly conservative (i.e., likely an overestimate) since it assumes that all errors occur in the same direction; in reality, digitisation errors are likely to produce both underestimates and overestimates of fast ice extent in equal measure. Furthermore, cyclonic systems which may cause wind-blown regional fast ice breakout

(Massom et al., 2009) also typically bring extensive cloud cover. In this way, image subsets requiring manual fast ice edge delineation are more likely to be produced during times of wholesale ice edge change, thereby falsely inflating the uncertainty estimates.

Regarding the fast ice dataset product, we provide the method of edge determination ("automatic" or "manual") in the output dataset, for each pixel of fast ice edge. We also compute the mean percentage of automatically-determined ice edge pixels in each 1° longitude bin. As a further indication of dataset integrity, we quantify differences between the new fast ice dataset and the Fraser et al. (2012) East Antarctic-only dataset for the period and region of overlap (10° W to 172° E, north of 72° S, March 2000 to December 2008). Large tabular icebergs are removed from the fast ice classification where independent iceberg information is available and/or the icebergs are clearly visible, but manual discrimination between fast ice and large tabular icebergs is difficult at times due to a lack of contrast in the satellite imagery (Fraser et al., 2010). Similarly, myriads of small icebergs embedded/grounded in places in the fast ice (Massom et al., 2009) are difficult to distinguish and remove, but form an integral part of the fast ice matrix. Following Fraser et al. (2010), we classify such regions of fast ice containing many small grounded icebergs as fast ice. Regions of ice mélange at the front of ice shelves are another source of uncertainty here, but remain unquantified due to their negligible areal extent on a continental scale.

# 3 Results and brief discussion

We restrict our presentation of results to illustration of the key attributes of this new pan-Antarctic fast ice dataset, and evaluate its improvements over earlier datasets created for East Antarctica (Fraser et al., 2012). We also present quantification of uncertainties. More in-depth analysis of spatio-temporal patterns and drivers of fast ice distribution is outside the scope of this manuscript, but is underway for future studies.

## 3.1 Circumpolar distribution of fast ice at maximum and minimum extent, and cross-comparison with earlier work

We illustrate the envelope of circum-Antarctic fast ice extent throughout the 18-year dataset time series by showing its spatial distribution at maximum (occurring in 2006, at DOY 271-285) and minimum (2009, DOY 061-075) extent in Figure 3. Figure 4 then shows a cross-comparison of this dataset with that of Fraser et al. (2012), covering the area and period of overlap. The total East Antarctic fast ice extent in the new dataset is 8.3 % greater than that reported in Fraser et al. (2012), on average. This difference is attributed to two factors: 1) a "relaxation" of the temporal fast ice condition in the new algorithm from the 20 day criterion used in Fraser et al. (2012), i.e., more ice remains "fast" for 15 days than for 20 days; and 2) the enhanced ability of the new "persistence of edges" algorithm to retrieve fast-ice extent under cloud cover. The largest differences between the two datasets are encountered at ∼118° E and 152° E. These two longitudes correspond to areas of dynamically-formed "semi-fast ice", i.e., regions where pack ice is blocked from westward advection, and intercepted by upstream obstacles e.g., large grounded iceberg B9B prior to its ungrounding in 2010 (Massom et al., 2010). In such regions, fast ice tends to be more exposed and ephemeral i.e., it can intermittently break out to become pack ice but then reform, on a synoptic scale. As such, reducing the temporal "fastness" condition to 15 days produces relatively large differences in these regions.

This sensitivity of fast ice extent to observation time-step has implications not only for the current work, but also for the next generation of SAR-based observations of fast ice, which, depending on the algorithm, can rely on two observations obtained in subsequent repeat passes. In the case of ESA's SENTINEL-1, this involves a 12 day repeat cycle. The temporal baseline of DLR's TerraSAR-X is shorter still at 11 days, although it has yet to be exploited for fast ice retrieval. Other SAR-based fast ice

retrieval algorithms which don't rely on exact repeat orbits are able to retrieve fast ice extent over even shorter baselines (e.g., feature-tracking algorithms can deal with any baseline, as long as features are present). Such methods are all likely to retrieve higher fast ice extents than the product here, simply due to the shorter observational baseline. As indicated here, differences are particularly strong in regions containing volatile fast ice. As such, end-users of fast ice products in such regions should be cognizant of this phenomenon.

**3.2   Quantification of dataset objectivity and error estimation**

Both the cloud-free composite images and the automated classification base images are susceptible to a number of factors which can reduce their quality/utility as fast ice edge discriminators. These include: 1) persistent/heavy cloud obscuration of the surface – particularly during times of no solar illumination when the cloud mask product is less accurate (Ackerman et al., 2006); and 2) instances where moving pack ice is advected toward the fast ice edge, thereby reducing the fast ice-pack ice

contrast in both visible and TIR images, as noted in Fraser et al. (2009).

Manual delineation ranges from being relatively straightforward (in the case of high quality composite imagery, where few judgement-calls need to be made) to quite labour intensive (in the case of heavy cloud obscuring the surface, resulting in ambiguous fast ice edge delineation, and requiring the use of the previous and next 15 day period's composite imagery for guidance). On occasion, such judgement-calls have the potential to significantly impact a single period's fast ice extent

retrieval, albeit in a limited region.

A broad measure of objectivity in fast ice extent retrieval is the percentage of edges that could be retrieved automatically. This is plotted in Figure 5. The circum-Antarctic mean automation percentage is 58%. East Antarctica is characterised by generally high automation percentages ($\sim$50 to 90%) – with the exception of localised pockets (down to 37%) located in Wilkes (98° to 108° E and 126° to 138° E) and George V lands (150° to 153° E). In West Antarctica, automation percentage is high (generally

70 to 90%) in the eastern Weddell Sea and Ross Sea (50 to 85%), but low in the Bellingshausen and Amundsen seas sector (40 to 60%) and along both flanks of the Antarctic Peninsula (as low as 22%). By showing longitudes with a low automation fraction, this plot also indicates areas which tend to be most affected by inherent issues detailed in the Methods Section, i.e., persistent cloud cover and/or persistent advection of pack ice toward fast ice that reduces the contrast (in reflectance and surface temperature) between pack and fast ice.

We have taken steps to mitigate this here compared to our earlier work (e.g., by now considering edges visible even under thin cloud; by more intelligently selecting the least-cloudy MODIS data for each 15 day period). Here, our approach is still limited by relatively poor MOD/MYD35 cloud mask product accuracy at times. In the future, we are interested in implementing state-of-the-art machine-learning cloud masking algorithms to mitigate this (e.g., Paul and Huntemann, 2020). This improvement may lead to an automation percentage in excess of the 58% reported here.

As detailed in the Methods Section, we estimated the sub-pixel error, applicable to both automatically- and manually-determined edges, as well as the manual-only error in digitisation. By simulation, the sub-pixel error is determined to be 0.288 pixels. We developed a novel technique to quantify the error in manual estimation of fast ice edges. We find that, on average, manually-determined edges change in location by 5.47 pixels more than that for automatically-determined edges (auto-determined = 10.06 pixels vs manually-determined = 15.53 pixels) in subsequent 15 day windows. Thus, the automatically-

determined edge error is 0.288 pixels, and the manually-determined edge error is the quadrature sum of 0.288 and 5.47 pixels, i.e., 5.48 px. For each 15 day epoch, we obtain a conservative estimate of the fast ice areal uncertainty by multiplying each skeletonised edge pixel by the appropriate error estimate, in km, assuming that the nominal resolution of 1 km/pixel applies everywhere in the domain. This uncertainty in fast-ice area has a mean value of 7.8% when averaged across the entire circum-Antarctic dataset. This is somewhat larger the value of 4.38% uncertainty obtained in regions requiring >10% manual edge

delineation, as detailed in Figure 5 from Fraser et al. (2010) using traditional re-digitisation-based error estimation, confirming that the new method is conservative.

## 4   Summary

Here we have both introduced: 1) a new improved technique for mapping and monitoring coastal fast ice coverage around Antarctica at high resolution, and 2) the most complete time series of Antarctic fast ice extent to date. This product represents

a new baseline against which to gauge change and variability in both the ice and climate, and has wide applicability. Indeed, it is expected to generate and contribute to multiple cross-disciplinary studies of the Antarctic coastal environment. Examples include behavioural ecology of charismatic megafauna (e.g., emperor penguin colony presence/absence), the effects of fast ice on the physical oceanography of the continental shelf (e.g., influencing coastal polynya location, and subsequent sea ice production and water mass modification), and a quantification of the fresh water, nutrients and biomass within the fast ice itself.

Logistical uses are also envisioned (e.g., informing base resupply schedules). Moreover, this dataset directly addresses a key gap identified in major high-level IPCC reports, enabling improved analysis of trends and variability of this key element of the highly-vulnerable Antarctic coastal environment (Vaughan et al., 2013; Meredith et al., 2019).

    The new algorithm also provides an important means of mapping and monitoring fast ice into the future and in a continuous fashion, given its applicability to the new generation of medium-resolution spectroradiometers. These include the Visible

Infrared Imaging Radiometer Suite (VIIRS) on NASA's Suomi National Polar-orbiting Partnership (NPP) platform (launched October 2011); the Sea and Land Surface Temperature Radiometer (SLSTR) and Ocean and Land Colour Instrument (OLCI) on ESA's Sentinel-3 platform (launched February 2016); and the Second-generation Global Imager (SGLI) on JAXA's Global Change Observation Mission (GCOM)-C1 platform (launched December 2017).

    Although an element of subjectivity remains in the large-scale retrieval of fast ice coverage from satellite visible/thermal

infrared imagery, we have mitigated this to some extent. This has been achieved by: 1) implementing an automated ice edge-retrieval algorithm, resulting in successful extraction of ∼58% of ice edge pixels; 2) performing random manual extraction to eliminate false trends; 3) quantifying the uncertainty associated with manual edge delineation (7.8 % of fast ice area retrieval,

on average); and 4) performing a cross-comparison with a similar (but independent) spatially- and temporally-overlapping dataset (Fraser et al., 2012). Crucially, this new MODIS-based dataset provides the longest contiguous time series of this key element of the Antarctic cryosphere while offering complete circum-Antarctic coverage for the first time at high resolution.

Multi-sensor fusion would help to further mitigate the subjective elements of this dataset to some extent. As an example, we used AMSR-E, in our previous work (Fraser et al., 2010). However, mission overlap generally limits the time period able to be considered in multi-sensor fusion algorithms (e.g., AMSR-E was launched 2.5 years after Terra MODIS, and was effectively decommissioned in 2011).

Analysis of spatio-temporal patterns, variability and trends in circum-Antarctic fast ice coverage is underway, using this dataset (Fraser et al., in prep.), as is related work determining and evaluating the drivers of these observed patterns. Moreover, we plan to study the spatial distribution of fast ice extent in the context of a new dataset describing the multiscale complexity and configuration of the coastline (including aspect) around Antarctica (Porter-Smith et al., in review, 2019), under the hypothesis that the coastal configuration is a first-order determinant of fast ice extent in many regions.

## 5   Data availability

The dataset has been made available at the Australian Antarctic Data Centre at http://dx.doi.org/doi:10.26179/5d267d1ceb60c, as a series of Climate and Forecast (CF)-compliant NetCDF files (Fraser et al., 2020). This dataset contains the following fields:

– Fast ice time series - presented as classified maps of the surface type (fast ice interior pixel; automatically-determined fast ice edge; manually-determined fast ice edge); and

– Latitude, longitude and area of each pixel.

There are plans to regularly update and extend the time series forwards in time, on a biennial basis, until the demise of both MODIS platforms but continuing with next-generation imaging spectroradiometers after this time.

*Author contributions.*   ADF led the study, acquired the data, developed automation algorithms, manually digitised fast ice, produced figures, and wrote the manuscript. RAM and KIO contributed equally toward project genesis and direction. SW contributed to algorithm automation development. PJK assisted with manual digitisation. JC and RP-S packaged the dataset for distribution. All authors edited the manuscript.

*Competing interests.*   The authors declare that they have no conflict of interest.

*Acknowledgements.*   MODIS data were obtained from the NASA Level-1 Atmosphere Archive & Distribution System Distributed Active Archive Center at: https://ladsweb.modaps.eosdis.nasa.gov. This work was supported by the Australian Government's Cooperative Research

Centre program through the Antarctic Climate & Ecosystems Cooperative Research Centre; the Japan Society for the Promotion of Science Grant-in-Aid for Scientific Research (KAKENHI) numbers 2503748, 24810030, 25241001, 26740007 and 17H01157; by the Canon Foundation; the Natural Environment Research Council (Grant NE/L002531/1); and the Australian Research Council's Special Research Initiative for Antarctic Gateway Partnership (Project ID SR140300001). This work also contributes to Australian Antarctic Science Project 4116, the Australian Antarctic Program Partnership and the World Climate Research Programme (WCRP) Climate and Cryosphere (CliC)

Project initiative *Interactions Between Cryospheric Elements*. RM acknowledges the support of the Australian Antarctic Division. ADF is grateful to D. Fanning for maintaining the Coyote library of IDL routines, many of which were used in this work, to T. Haran (NSIDC) for developing the MODIS Swath-To-Grid Toolbox, and to C. Greene for discussions around the distribution of the data. The authors would like to thank the four anonymous peer-reviewers for their careful and constructive criticism of the originally-submitted manuscript.

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

**Figure 1.** Flow chart depicting the image processing pipeline. Bold letters within the green-coloured elements refer to individual panels in Figure 2.

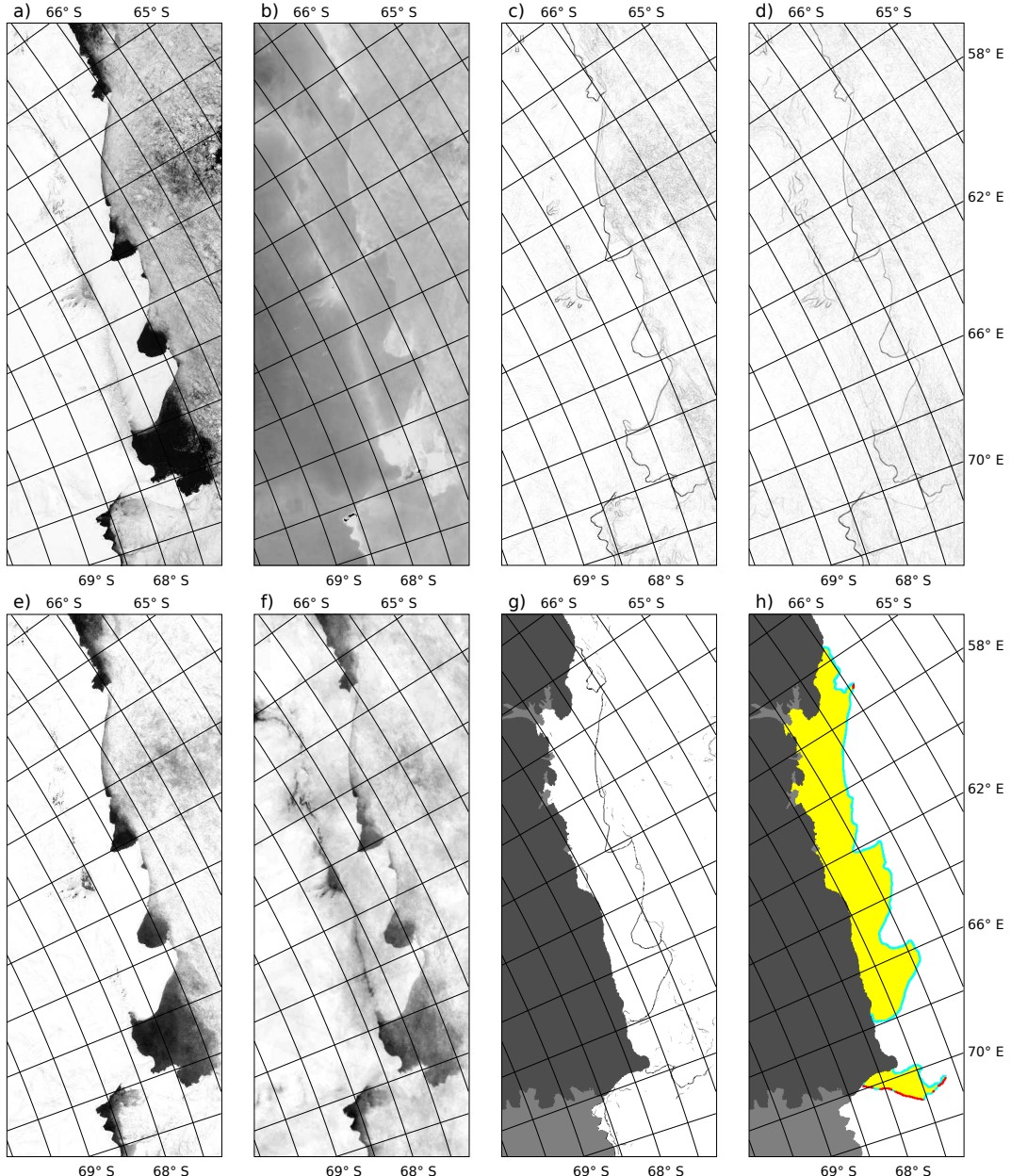

**Figure 2.** Figure depicting an example of the automated fast ice edge detection along the Mawson Coast, East Antarctica, for DOY range 316-330, 2005. See the red rectangle in Figure 3 for spatial context. a) and b): 15 day channel 1 (visible) and channel 31 (thermal infrared) cloud-free composite images, respectively. c) and d): Sum of Canny algorithm-detected edges in individual channel 1 and channel 31 images respectively, for the 15 day period. e) and f): Modified lead-detection for channel 1 and channel 31 images, respectively (after Willmes and Heinemann, 2015, but with an enlarged filtering window to enhance fast ice detection). g) Results of the combined edge detection algorithm (black line). Light and dark grey areas represent grounded and floating glacial ice, respectively, and are masked out. h) Fast ice classified map after manual edge inspection/correction and filling. Cyan and red represent automatically- and manually-completed edges, respectively, and the width of these lines has been expanded in this example to enhance visibility. Yellow represents infilled fast ice area.

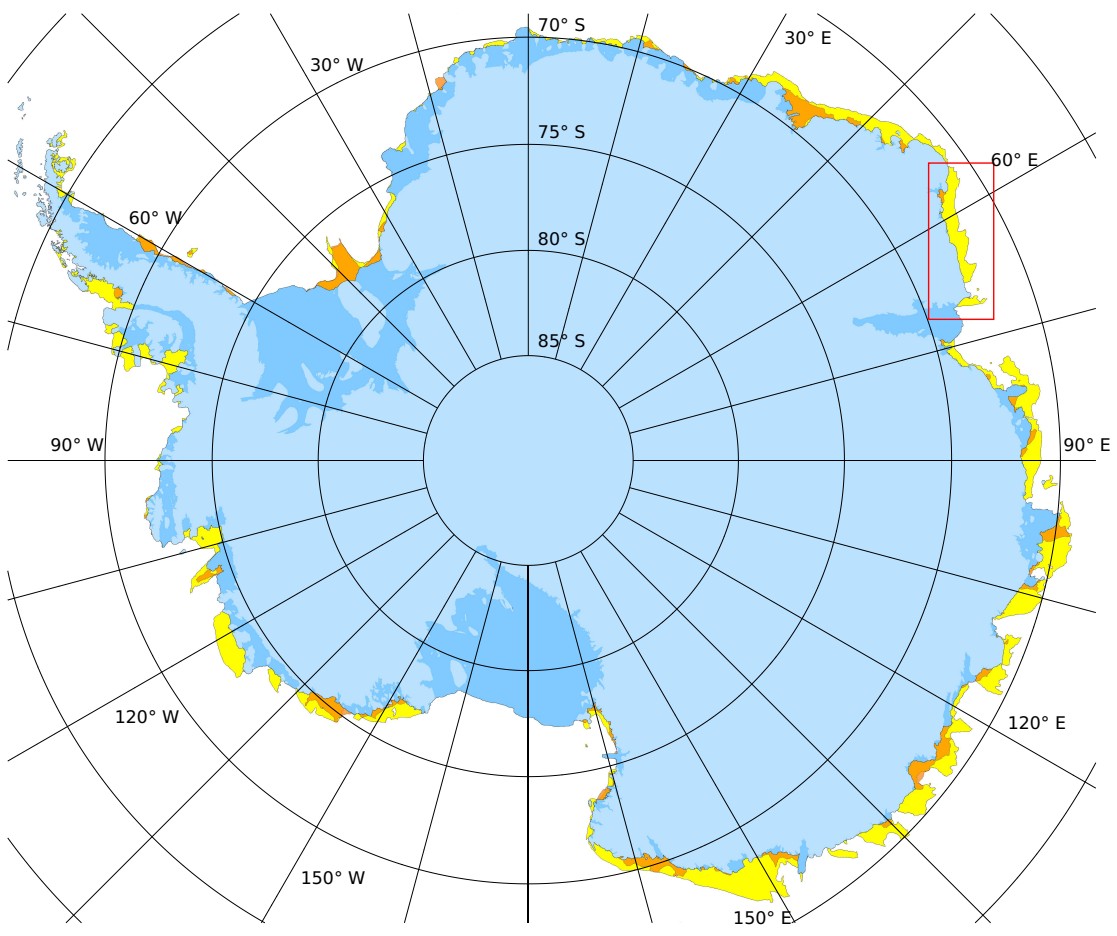

**Figure 3.** Fast ice distribution at times of maximum (occurring in 2006, DOY 271-285; shown in yellow) and minimum (2009, DOY 061-075; shown in orange) extent over the 18 year dataset period. The grounded Antarctic Ice Sheet and floating ice shelves are shaded light and dark blue, respectively. The red rectangle shows the region used to illustrate the automation in Fig. 2.

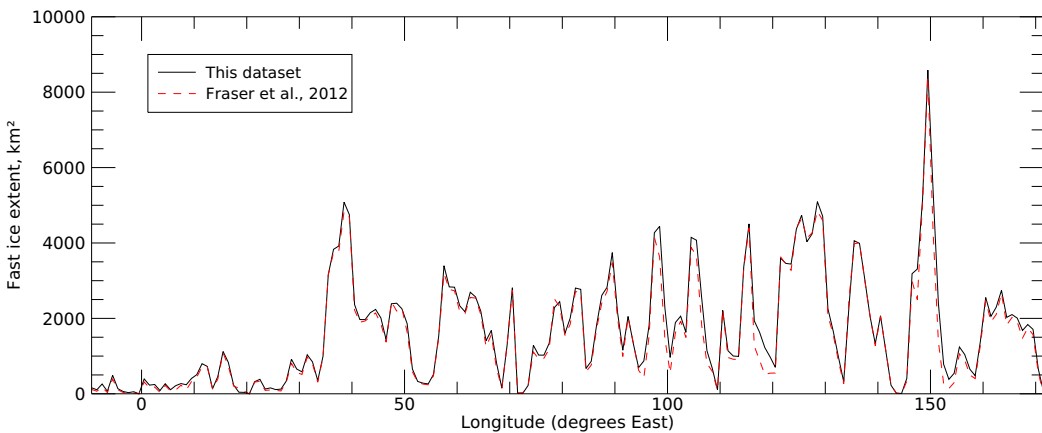

**Figure 4.** Mean fast ice extent per degree of longitude for this new improved dataset (black solid line) and Fraser et al. (2012) (dashed red line), for the period and region of time series overlap (March 2000 to December 2008, 10° W to 172° E).

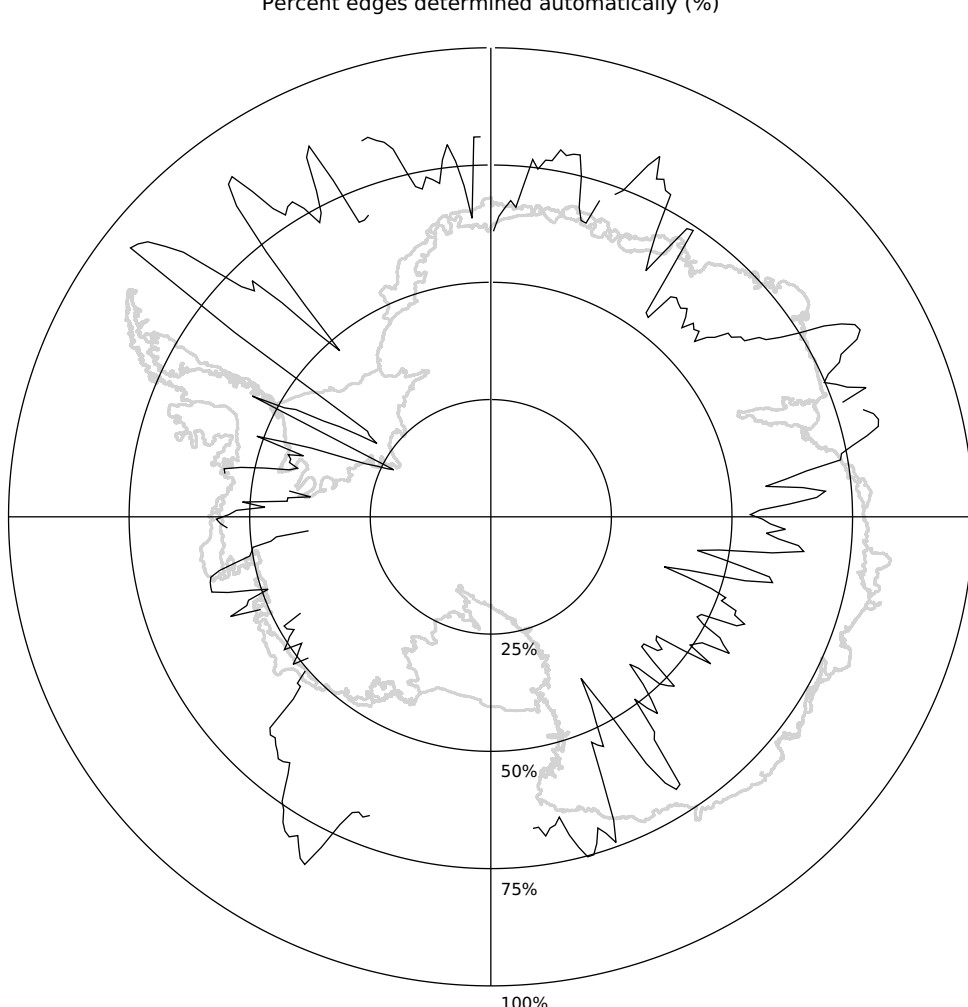

**Figure 5.** Polar plot showing the percentage of edges determined automatically, as a function of longitude. The Antarctic continent is outlined in grey for spatial context. To remove noise in regions with little fast ice, 1° longitude bins with less than 5,000 total fast ice edge pixels across the 18 year dataset were not plotted.