# Peer review of "High-resolution mapping of circum-Antarctic landfast sea ice distribution, 2000-2018"

_Earth System Science Data, 2020_

## Referee Comment (RC1) · Anonymous Referee #1 · 30 Jun 2020

Dear authors,

I am pleased to review this manuscript as this is a most relevant and timely paper as it sheds light on and important topic as the authors points out. Much work has been attributed to the landfast ice in the Arctic, but less so in the Antarctic. The paper presents a pan-Antarctic dataset and preliminary analysis of the landfast ice extent in the Antarctic from 2000 to 2018. This is a novel effort and I recommend this for publication in The Cryosphere. Furthermore, the manuscript is well prepared as it is concise and well written. In particular the introduction, which is straight to the point and encompasses the relevant literature.

I mostly have minor comments and suggestions, but would like to see two main issues being addressed prior to publication related to the description of methods as well as discussion of uncertainties. In short, I think the first part of the methods should be clarified better and elaborated on. The second part of the methods should be moved to a separate subsection or preferably a discussion section with some more discussion about accuracy and caveats.

With best regards,
Your reviewer

**Description of methods:**

I am not confident I understand what you have done strictly based on this manuscript. As I think it would be good if this paper stands by itself, I am suggesting that you spend some more time elaborating on the methods.

For instance, it is not clear to me how you create the composites. Could you be clearer in terms of this being a mosaic of overlapping imagery or if you are considering just one acquisition for each location and this being a composite of the two channels. Either way, it would be helpful with a couple of sentences addressing how they are created and the number of images typically incorporated.

The steps in the methods are quite clear in of themselves, but it would help if the outcome of each steps is described as well. This is how I interpret your initial steps:

- For any given location and for all 15-day periods in your dataset, you download all available images and create multiple composite images based on available channels (how do you do this?). Hence, for this location, you have several images (how many roughly?)
- Then you detect the edges in all these composites resulting in multiple 1 km resolution (binary datasets?) indicating locations of edges in each composite.
- Then you sum the binary datasets and thus higher numbers more strongly indicate persistent edges at the timescale of 15 days. By doing this you reduce the number of composites down to one product?

- You then multiply the edges with the median-filtered composite. But which one, if you have multiple composites for this region? Also, please spend a sentence on describing how this results in confidence as opposed to just the edge product. What is the range of values prior to normalization?
- Finally, the result is a product of landfast ice edge with 1 km spatial and 15-day temporal resolution. However, how do you eliminate lower confidence edges based on the histogram analysis?

If you could attempt to clear up any of misunderstandings and make this a little easier to follow, it would be great. I suggest a figure where the reader can associate each step with a figure panel (or alternately a schematics). Basically, modifying Figure 1 to incorporate the other steps as well.

**Accuracy estimates:**

At line 190 you describe: "In the case of a manually-extracted ice edge pixel, it reflects the sum of the ice edge change plus the digitisation error." And you seem to imply that there are no errors in this data? Could you elaborate about this in the manuscript and discuss the accuracy of the Canny edge detection in general based on your pixel spacing and in terms of misclassification in the case of slow moving non-stationary ice in fjords and stationary drifting ice pinned between icebergs or by onshore winds over consecutive 15-day periods? I suggest as before to move some of this discussion into either a discussion section or as a subsection to the methods section and discuss caveats a little more deliberately.

You define landfast ice as stationary for 15 days as opposed to earlier 20 days. A 3-week timeframe is to my knowledge more common. Why did you make this choice and how will this impact the analysis and the potential misclassification of temporarily stationary pack ice etc.

Is the manual delineation very labor intensive e.g. is it sometimes difficult to determine where to draw the line with potentially large consequences for the ice extent? Do you have suggestions for how to mitigate this or how your approach could be improved in the future resulting in a larger than 58% success rate? If you could discuss this slightly in the manuscript, that would be very interesting. Could you even provide some speculation into whether other sensors could enhance the analysis?

I realize and appreciate that you have written a quite concise paper and don't want to delve too much into the details. However, I suggest some more clarity and elaboration around these two points.

Minor comments and questions

Line 65 - 74 move to discussion
Line 78: I don't see the "in prep" reference in the bibliography. If not included there, take out.

Table 1: Avoid the word very as in "very high"

Line 81: Replace "the new" with something like "the fast ice time series presented here" to make it clear that it is not a new one described in Table 1.

Line 95: Can you clarify how the composites are created? Do you mean creating a mosaic or merging the channels?

Line 97: Is this manual updating done every year and for the entire coastline? Is this labor intensive if to be done with necessary accuracy?

Line 112: "layer of clouds"?

Line 115: Here as well as prior in the manuscript, the use of parenthesis could be toned down by reformulating.

Line 123: Again, not sure if parenthesis is needed here.

Line 126: I am not familiar with the plural form "cloud". Clouds?

Line 125: You have already said this. Take out.

Line 133: Missing oxford comma

Line 139 and 140: It would be great if you could elaborate here on what you mean by summing edge products. Do you just sum binary pixel values of edge/no edge?

Line 139: Do you mean successive 15-day periods, meaning several periods? If so, how many?

Line 142: Try to limit redundancy, you have already stated that the composites are cloud-free

Line 143: Could you provide a short explanation for the Median filter. For instance, what is this gradient value range? Is there a threshold used to determine whether the edge is stationary and for how long?

Line 148: Is this something you define. If so, make that clear. Otherwise, please provide reference.

Line 162: In the methods section below Line 162 looks like the start of a discussion to me. I recommend creating a discussion section and placing much of this there. Some of it also belongs more in the introduction perhaps.

185: Like before, no need for parenthesis

Line 215: Missing space

Line 244: What indicates this in the plot? The discontinuity in the plot?

Line 248: What do you mean with edges vary? The detected edges or the actual ice edge? You mean vary over time when the ice edge is assumed constant? Please explain better.

Line 263: Please be consistent with the use of notations for in-line lists e.g. 1, a, or i.

Line 266: Missing space

Line 271: Please clarify this sentence as it is not clear what you mean by complexity dataset and linkages between what.

---

## Referee Comment (RC2) · Anonymous Referee #2 · 6 Jul 2020

To the Authors,

This is an interesting and relevant study that advances a new dataset as well as preliminary analysis on Antarctic landfast ice conditions from 2000-2018. The topic is appropriate for this journal, and is timely given increasing interest in sea ice change and its consequences. The study also fills an essential knowledge gap in landfast ice studies, which tend to focus on Arctic and Subarctic environments, by introducing a pan-Antarctic dataset to serve as the basis for future research on a relatively understudied continent. I recommend this work be published after some minor revisions which are listed below. I first provide general feedback, followed by line-by-line revision requests. Feedback generally concerns editing with the writing and clarification on data properties. I understand the methodology to be a modification on already-established

approaches advanced in previous publications and accept this as valid.

General notes:

I think the spatiotemporal dimensions of the landfast ice datasets you create (1km, 15-day interval) could be better justified. It's entirely reasonable to cite the data product and repeat interval of the satellite as reasons for these dimensions. However, because the purpose of this publication is to present a novel dataset for others to use, it would be a good idea to include some discussion of the advantages/disadvantages of these spatiotemporal dimensions. I understand you are intending to apply this dataset in an analysis for future publication. I would advise you either discuss how these dimensions apply to your intended use of the dataset, or how you envision others using your dataset. In your results section, for example, you observed an 8.3% increase in fast ice extent compared to Fraser et al.'s (2012) study, which you attributed to the switch from a 20 to 15-day stationary criterion used to identify fast ice. How do these differences in outcomes due to changes in temporal window affect what this data might be used for?

Line-by-line revision requests:

Line 67: Perhaps provide some examples of these scientific and operational uses?

Line 81: The use of parenthesis to clarify the imagery dating back to the year 2000 seems out of place. It would help the flow of the introductory sentence to this section to find a way to integrate this into the sentence without the use of parenthesis.

Line 96: I'm unsure of why "(coastline)" and "(change in)" were inserted into this sentence. Please edit the sentence to make their purpose clear.

Line 98: The wording of this sentence is a little redundant. You can change it to "Temporal compositing was carried out to create cloud-free..." or "Temporal compositing is required to create cloud-free...". Since this is the methods section, the reader will already assume this is what you've done, so I would recommend the latter. It is also in keeping with the present tense used in the writing.

Line 106: Because the authors are the same for the two studies cited, you can change the in-parenthesis citation to Fraser et al. (2009, 2010).

Line 107: If you are referring to both cited Fraser et al. studies, say "The earlier works", if you are referring to only one of the cited studies, specify which one.

Line 110: Per my comment on line 107, if Fraser et al. 2010 is the work being referenced, make mention of it earlier. Perhaps move this parenthesis citation to the end of the previous sentence that starts with "The earlier work".

Line 111: Overall the writing in this manuscript is well done. However there is the tendency to use parenthesis when they are not necessary. The clarification that cloud cover is a challenge for optical remote sensing in polar regions can either be integrated into the sentence, or stand as its own sentence. I would recommend the latter, as this would allow for the inclusion of study citations where optical remote sensing was challenging in polar environments.

Line 118: Please integrate parenthesis into sentence, or create new sentence.

Line 123: Please integrate parenthesis into sentence.

Line 126: Please change to "thin clouds" or "thin cloud cover".

Line 168: Figure 1 is really well done, and does a good job complimenting the written description of the data collection process.

Line 182: Parenthesis integrate or remove

Line 187: Parenthesis integrate or remove

Lines 192 - 205: I am personally a supporter of numbered lists in publications, especially methods sections, as they are a great help for the audience. However I would recommend some consistency in how these numbered lists are used. From lines 192 - 205 there are two numbered lists, the first list is independent from the text in a line-by-line format. The second list is integrated in the text. I advise you pick a method of

numbering and stick with it. If you choose to integrate both lists into the text, I suggest you separate them into different paragraphs so the readers do not get them confused.

Line 215: Add a space between the final word and the parenthesis containing the citation.

Line 220-223: This sentence needs work. I would advise either choosing between "ground-breaking" and "new" to avoid redundancy. Remove the parenthesis (across East Antarctica) and integrate into text. Rearrange to improve the flow. For example: "We restrict our presentation of results to the illustration of key attributes in this new pan-Antarctic fast ice dataset, and evaluate its improvements over earlier datasets created for East Antarctica (Fraser et al. 2012).

Line 223-224: Remove parenthesis and integrate into text, or delete it. In this case I would advise the latter because the audience already knows you are talking about the dataset you created. Also, the comma separation breaks the flow of the sentence. Try something like: "More in-depth analysis of spatial-temporal patterns and drivers of fast ice distribution is outside the scope of this journal, but is underway for future studies (Fraser et al., in prep.)."

Line 227: I would remove "important new" adjectives in this sentence when you are referring to the data. The importance has already been demonstrated in the intro and discussion sections, and the purpose of the article is to introduce a novel dataset, so the audience already knows it is new.

Line 230: Please integrate the parenthesis text into the sentence.

Line 233: Please integrate the parenthesis text into the sentence

Line 236: Please integrate the parenthesis text into the sentence

Line 259-260: Can you specify any ongoing or anticipated study topics of the Antarctic coastal environment this dataset is expected to help? It's okay if there aren't any that can be specified at the present time, but it would be interesting to include if there are.

Line 260: As I make clear in my summary of this manuscript above, I have no doubt this dataset is a very important contribution to Antarctic fast ice research, and will be heavily cited for years to come. However there is a certain promotional tone in this manuscript that seems out of place in a scientific article. In line 260, "major high-level" is used to emphasize the importance of IPCC reports. However, readers of ESSD will already know the importance and weight of IPCC reports, and will not need these adjectives. Unless "major" and "high-level' are established terms used to organize IPCC reports by importance, I would advise leaving them out. Throughout the manuscript you qualify mentions of your data with terms such as "new" and "ground-breaking". While this dataset is indeed new and ground-breaking, it would be better to reserve these terms for sentences when the actual importance of the data is directly addressed, rather than somewhat indiscriminately throughout the manuscript. I understand the purpose of this paper is to make the availability and utility of this new dataset known to the scientific community. I would argue, on your behalf, that the importance of the dataset you created is already evident in your paper, and the scientific community will readily understand this without the need for promotional adjectives.

Lines 263-267: Previously you used numbers when listing steps taken to accomplish a goal. I suggest using numbers here instead of letters, to maintain consistency in the paper.

Line 269: In this paper you use the terms "spatial-temporal patterns" and "spatio-temporal patterns". Both are valid terms, but for the sake of consistency I would pick one and use throughout.

---

## Referee Comment (RC3) · Anonymous Referee #3 · 21 Jul 2020

This manuscript presents a novel algorithm for determining pan-Antarctic fast ice area over 15-day epochs. It has been applied to an 18-year record of MODIS imagery to produce the first pan-Antarctic dataset of fast ice edge area. It marks significant improvements over existing algorithms that were regional in nature and required more manual interpretation to define fast ice edges. This is timely and important work as it fills a significant gap in the knowledge of Antarctic fast ice area variability. This will benefit both scientific investigations and logistical operations in key areas of the Antarctic coastal environment.

The manuscript is well-written and concise, and I recommend it for publication in Earth System Science Data with minor revisions. I have divided my comments into three sections; general comments on the manuscript, minor comments on the manuscript

and comments on the data set.

General comments on the manuscript

I found the description of the algorithm in the methods section somewhat difficult to follow. I would recommend creating a flow-diagram to better illustrate how the algorithm is applied in general This diagram could then refer to Figure 1 to illustrate outputs at various steps in the algorithm. I would also like to see more detail on some aspects of the algorithm. For example, how does the algorithm deal with cases where both thermal and visible imagery are available when generating the 15-day cloud-free composite images? I would also like to see some discussion in the results section on whether there were observed differences between fast ice area products generated from visible and thermal composite images. Further, I would like to see more justification for choosing a 1-km, 15-day epoch for identifying landfast sea ice, and more discussion on how the choice of this epoch influences the generated fast ice extent products.

I would also like to see more discussion on the fast ice distributions shown in Figure 2. Antarctic fast ice extent can be temporally variable on a regional scale, and I would argue that this variability is not captured by presenting pan-Antarctic maximum and minimum distributions. For example, the fast ice edge in McMurdo Sound in 2016 was significantly farther from the coast than shown in Figure 2 (see, for example MYD02.A2016350.0410.006).

The authors state that the number of images contributing to the composite was increased relative to the Fraser et al. (2019) algorithm (Lines 114 + 115). I would like to see more details on how this was accomplished, particularly since the epoch was reduced from 20 to 15 days. If I understand correctly, the auto-determined fast ice edge moved an average of $\sim 10$ km in a 15-day period. How does this compare to previous regional studies?

The authors state that four adaptive thresholds are set when computing fast ice edge confidence, but then do not describe how these thresholds are utilised in the algorithm.

Please provide this detail.

Minor comments on the manuscript

Line 7: visible-thermal infrared imagery – change to "compositing visible and thermal infrared imagery".

Line 38: change ", but at a poorer spatial resolution of âĹij6.25 km (Nihashi and Ohshima, 2015) to limit its" to ", but a poorer spatial resolution of âĹij6.25 km (Nihashi and Ohshima, 2015) limits its"

Lines 65 – 75: this would fit better in the results section.

Line 66: suggest re-order "It also has a multitude of potential scientific and operational uses, given the wide-ranging importance of fast ice" to "Given the wide-ranging importance of fast ice, it also has a multitude of potential scientific and operational uses."

Line 68: remove "developed"

Line 95: Can you estimate how time intensive it is to update the coastlines and ice shelf edge positions on an annual basis?

Line 96: it is not clear what is meant by "change in".

Line 104: where are the data provided?

Line 139: what is meant by "successive"?

Line 139 + 140: provide more detail by what is meant by "sum over".

Line 142: Provide more detail on how the absolute value of the gradient for the composite image was calculated.

Line 149: remove "are set".

Lines 154 – 158: how time intensive is it (on average) to undertake manual processing of fast ice edges? How are the lead-detection images used in the manual processing?

Line 178: replace "Here and" with "Here, "

Lines 195 + 197: provide more detail on how the mean fast ice edge separation between composite subsequent images is calculated, e.g. how do you determine which pixel in the second image to "match" with the pixel in the first image?

Line 202: explain what is meant by "... all remaining manually-determined pixels ..."

Line 223: replace "journal" with "manuscript".

Line 248: confirm whether the time period over which these variations have been calculated is 15-days.

Comments on the data set

In the data set's README file, it states that the latitude of true scale is 70 N. This should read 70 S.

---

## Referee Comment (RC4) · Anonymous Referee #4 · 31 Jul 2020

This paper evaluates the detection of Antarctic fast ice variations, then is scientifically very good. I recommend the MS for publication in ESSD with minor revisions. The followings are minor comments: 1) How have authors distinguished between fast ice and ice shelves from satellites data? ãĂĂAs with fast ice, it is expected to occasionally collapse at the edge of the ice shelves. 2) Regarding the Figure 4, why are the solid line discontinuous in some places in this figure? Also, in the discussion section 3.2, the authors have described the results of the comparison between the East and West Antarctic regions. Is it possible to consider the reasons for the differences in terms of sea ice, ocean processes and/or atmospheric fields?
* * *

---

## Author Comment (AC1) · 19 Aug 2020

(Responses separated into general comments to Reviewer 1, "R1A" and "R1B" as main issues, then "1)", "2)", etc for specific minor comments)

General comments to Reviewer 1: We thank Reviewer 1 for their encouraging preamble and very careful review of the manuscript. We generally support all Reviewer 1's suggestions and recognise that they will improve the manuscript. All comments are specifically addressed below, and will be incorporated into the revised manuscript.

**R1A: Description of methods: - I am not confident I understand what you have done strictly based on this manuscript. As I think it would be good if this paper stands by itself, I am suggesting that you spend some more time elaborating on**

**the methods.**

We are happy to provide more details in this section. As you identified, we tried to make the manuscript as concise as possible, but acknowledge that more detail in this section would be a wise way to spend words.

**- For instance, it is not clear to me how you create the composites. Could you be clearer in terms of this being a mosaic of overlapping imagery or if you are considering just one acquisition for each location and this being a composite of the two channels. Either way, it would be helpful with a couple of sentences addressing how they are created and the number of images typically incorporated.**

One thermal infrared composite image was created (from Channel 31), regardless of the time of year. During times of sufficient sunlight, a visible (Channel 01) composite image was also created. So to answer your question, it is a mosaic of overlapping imagery – in fact, two mosaics of overlapping imagery in periods of sunlight (one for reflectance, one for brightness temperature). In line 132 we state that 600 images are incorporated into the composite images for each 15 day period, but we are happy to elaborate on this in the text by saying that these 600 images are separated into 6 regions of 100 images. Without this regional consideration, we found that there is a concentration of images in one or more particular regions based on cloud conditions, since we rank and select the 600 least cloudy granules.

**- The steps in the methods are quite clear in of themselves, but it would help if the outcome of each steps is described as well.**

This is a great idea, and will increase the clarity of explanation. We will add this in the revised document.

**- This is how I interpret your initial steps: - For any given location and for all 15-day periods in your dataset, you download all available images and create multiple composite images based on available channels (how do you do this?).**

[Figure]

Not quite, and sorry for the confusing/insufficient explanation. We download all (approx. 1,800 per 15 day period) cloud mask ("MOD/MYD35" product) granules covering the Antarctic coast. We then grid these, partition into six regions around the coast, and rank each group to select the top 100. For these 600 timestamps, we then download the calibrated radiance data ("MOD/MYD02") from which composites and the other processing steps are conducted. We will include this deeper description in the revised manuscript.

**- Hence, for this location, you have several images (how many roughly?)**

The composites for each region (of approx. 60 degrees longitude width) have 100 input images. Note that the "regional" consideration is dropped after the cloud mask product has been gridded, i.e., the MOD02 product is gridded to the circumpolar grid. Of course, these are not cloud-free views of the surface for all pixels of each MOD02 granule. Again, we are happy to include more detail on this.

**- Then you detect the edges in all these composites resulting in multiple 1 km resolution (binary datasets?) indicating locations of edges in each composite.**

That's correct – as mentioned in line 142, composites undergo edge-detection. However, edges are also computed for each input granule, as mentioned in lines 139 and 140, resulting in multiple 1 km resolution binary datasets indicating edges in each granule (not composite). Reviewer 3 suggested a flow chart. I think this would be a great complement to the description in the text and Fig 1, so we will produce this, and cross-reference Figure 1's individual panels from this chart.

**- Then you sum the binary datasets and thus higher numbers more strongly indicate persistent edges at the timescale of 15 days. By doing this you reduce the number of composites down to one product?**

That's right. Well, not quite one product, but we vastly reduce the input images by this process. We retain the Canny vs Sobel edges in separate summary images, for

example.

**- You then multiply the edges with the median-filtered composite. But which one, if you have multiple composites for this region?**

That's right. This is Step 6 detailed in line 146. Only the (summed) Canny edges are used in this process to construct the edge probability map. This is because the Canny edges have excellent localisation.

**- Also, please spend a sentence on describing how this results in confidence as opposed to just the edge product.**

That's a good idea. Since both the summed Canny input map and the gradient-median-filtered composites and non-binary, their product is also non-binary and so gives a fine-grained measure of confidence. Using four adaptive thresholds (based on the histogram of the value of the product of these two maps), we assign four broad confidence classes to edges.

**- What is the range of values prior to normalization?**

Good question. I haven't thought to check, because the normalised product is so much more useful when deploying this algorithm across the whole continent. The lower range is 0 (no Canny edges * a zero value for the median-filtered composite image). The upper range is typically the product of 60 Canny edges (a particularly obvious edge with frequent, clear views of the surface throughout the whole period in both Terra and Aqua MODIS) and a composite gradient-median-filtered value of around 5, for the infrared case, or 0.75, in the visible channel case. The IR value is higher due to the numerical value of the brightness temperature difference between cold ice and warm water being much higher than the difference in reflectance of a dark vs light surface. But these details are helpfully abstracted away thanks to the normalisation.

**- Finally, the result is a product of landfast ice edge with 1 km spatial and 15-day temporal resolution. However, how do you eliminate lower confidence edges**

**based on the histogram analysis?**

Edges with a lower confidence than the lowest threshold (which is 98

**- If you could attempt to clear up any of misunderstandings and make this a little easier to follow, it would be great. I suggest a figure where the reader can associate each step with a figure panel (or alternately a schematics). Basically, modifying Figure 1 to incorporate the other steps as well**

Thanks – this is a good suggestion. We also note that Reviewer 3 suggests a flow diagram. We think that a clearer explanation, as you outlined, plus the flow diagram will help greatly.

**R1B:**

**Specific points: -At line 190 you describe: "In the case of a manually-extracted ice edge pixel, it reflects the sum of the ice edge change plus the digitisation error." And you seem to imply that there are no errors in this data?**

I think you meant to type that this implies that there is no error in the automatic digitisation, is that right? This (that the auto-determined pixels have no error) was our initial assumption – since the Canny edge localisation is very good – however you're right - we have implicitly ignored any sub-pixel digitisation error with this assumption.

Performing a quick random point simulation, I can see that the sub-pixel error averages to zero, but has an RMS value of 0.288 px. It could be argued that this value is a better one to use here. In the revised manuscript we plan to incorporate this number into the error analysis.

**- Could you elaborate about this in the manuscript and discuss the accuracy of the Canny edge detection in general based on your pixel spacing and in terms of misclassification in the case of slow moving non-stationary ice in fjords**

Misclassification of melange as fast ice probably occurs in a few limited regions around

the coast. We haven't mentioned this in the manuscript because it probably occurs in such limited regions as to be negligible on a circum-Antarctic scale. But I agree that it's worth noting this caveat in the revised manuscript. Another related error is in regions of densely-packed icebergs, which we pointed out in line 215, so this would be a good place to discuss melange misclassification.

**- and stationary drifting ice pinned between icebergs or by onshore winds over consecutive 15-day periods? I suggest as before to move some of this discussion into either a discussion section or as a subsection to the methods section and discuss caveats a little more deliberately.**

Yes, as you indicate, both issues are probably present to some extent. Regarding the drifting sea ice pinned between grounded icebergs, we have experienced this but only in limited areas in the Antarctic (e.g., visible from stations but at a spatial scale much smaller than one km, our pixel size). Regarding the ice temporarily advected against the shore or existing, genuine fast ice, we still believe that 15 days is long enough to preclude most of this ice from consideration. The coastal flow is generally offshoreward to westward (Turner and Pendlebury, 2001). Blocking anticyclonic pressure systems do occur in southern midlatitudes and these can result in persistent onshoreward winds in particular regions of the Antarctic coast, although the residence time for such systems is rarely longer than one week (Massom et al., 2004). We plan to edit the text to discuss both caveats.

Refs:

J. Turner; S. Pendlebury (Eds.) The International Antarctic Weather Forecasting Handbook, British Antarctic Survey, Cambridge, UK. ISBN: 1855312212, 2001.

R. A. Massom; M. J. Pook; J. C. Comiso; N. Adams; J. Turner; T. Lachlan-Cope; T. T. Gibson. J. Climate (2004) 17 (10): 1914–1928

**- You define landfast ice as stationary for 15 days as opposed to earlier 20 days.**

**A 3-week timeframe is to my knowledge more common. Why did you make this choice and how will this impact the analysis and the potential misclassification of temporarily stationary pack ice etc.**

As above – the circumpolar trough generally permits swift passage of low-pressure systems from west to east. Blocking events can occur north of the ice edge, but these rarely persist more than one week, so 15 days is probably sufficient to exclude this except for extreme cases. Discussion to be added in the text.

**- Is the manual delineation very labor intensive e.g. is it sometimes difficult to determine where to draw the line with potentially large consequences for the ice extent? Do you have suggestions for how to mitigate this or how your approach could be improved in the future resulting in a larger than 58% success rate?If you could discuss this slightly in the manuscript, that would be very interesting.**

The manual delineation ranges from being relatively straightforward (in the case of high quality composite imagery, where few judgement calls need to be made) to quite labour intensive (in the case of heavy cloud obscuring the surface, resulting in ambiguous fast ice edge delineation, and requiring the use of the previous and next 15 day period's composite imagery for guidance). On occasion, such judgement calls have the potential to significantly impact a single period's fast ice extent retrieval in a limited region.

We have taken steps to mitigate this here compared to our earlier work (e.g., by now considering edges visible even under thin cloud; by including more MODIS data per 15-day period). Multisensor fusion would help alleviate this to some extent (we used AMSR-E in our previous work) but limits the time period able to be considered (e.g., AMSR-E was launched 2.5 years after Terra MODIS). Here, our approach is still limited by poor MOD35 cloud mask product accuracy at times. We are interested in implementing state-of-the-art machine-learning cloud masking to mitigate this (e.g., Paul and Huntemann, The Cryosphere Discussions, 2020). This improvement may lead to

an automation percentage in excess of the 58% reported here.

I agree that this kind of discussion is a great addition to the manuscript, and we plan to incorporate it in the revised version.

Ref: Paul, S. and Huntemann, M.: Improved machine-learning based open-water/sea-ice/cloud discrimination over wintertime Antarctic sea ice using MODIS thermal-infrared imagery, The Cryosphere Discuss., https://doi.org/10.5194/tc-2020-159, in review, 2020.

**- Could you even provide some speculation into whether other sensors could enhance the analysis?**

Yes – as above, AMSR-E has been used to complement this technique in our previous work (Fraser et al., 2010, which was purely manually-digitised), although it isn't clear how the multisensor fusion could be achieved in the framework of the present paper. Again, we would like to reiterate that few sensors match the very long observational lifetime of MODIS, so a multisensor fusion becomes less attractive in this sense.

**- I realize and appreciate that you have written a quite concise paper and don't want to delve too much into the details. However, I suggest some more clarity and elaboration around these two points.**

We agree that some more detail in these sections would be a useful addition at the cost of a few sentences.

**R1 - Minor comments:**

**1) Line 65 - 74 move to discussion**

Yes, I can see how it fits well in the discussion.

**2) Line 78: I don't see the "in prep" reference in the bibliography. If not included there, take out.**

Yes, this is still in prep so we will remove it from here.

**3) Table 1: Avoid the word very as in "very high"**

OK.

**4) Line 81: Replace "the new" with something like "the fast ice time series presented here" to make it clear that it is not a new one described in Table 1.**

Good idea.

**5) Line 95: Can you clarify how the composites are created? Do you mean creating a mosaic or merging the channels?**

Yes, addressed in the main comment above.

**6) Line 97: Is this manual updating done every year and for the entire coastline? Is this labor intensive if to be done with necessary accuracy?**

Yes, every year for the entire coastline, using the two MOA products (produced in 2004 and 2009) as a baseline. It is relatively quick in comparison to the manual parts of fast ice retrieval. Detail to be added to the text.

**7) Line 112: "layer of clouds"?**

OK

**8) Line 115: Here as well as prior in the manuscript, the use of parenthesis could be toned down by reformulating.**

Thank you – we note that Reviewer 2 has recommended a parenthesis overhaul too. We will revisit all occurrences in line with these comments.

**9) Line 123: Again, not sure if parenthesis is needed here.**

As above.

**10) Line 126: I am not familiar with the plural form "cloud". Clouds?**

Yes, will be changed as with comment 7).

**11) Line 125: You have already said this. Take out.**

Thank you.

**12) Line 133: Missing oxford comma**

Will be inserted.

**13) Line 139 and 140: It would be great if you could elaborate here on what you mean by summing edge products. Do you just sum binary pixel values of edge/no edge?**

That's correct. Detail to be added to the text.

**14) Line 139: Do you mean successive 15-day periods, meaning several periods? If so, how many?**

Sorry, this sentence was explained terribly! Thank you for picking it up. Will be changed from "and sum over successive 15-day periods." to "and summed within the current 15-day period." Same for the following dot point.

**15) Line 142: Try to limit redundancy, you have already stated that the composites are cloud-free**

Thank you, will be changed.

**16) Line 143: Could you provide a short explanation for the Median filter. For instance, what is this gradient value range? Is there a threshold used to determine whether the edge is stationary and for how long?**

Partly addressed in the response to your main comment above, but to explicitly answer here: The gradient of the median of the composites ranges from 0 to around 5 (for the Channel 31 thermal IR brightness temperature composite) or 0.75 (for the Channel 01 reflectance composite). There is no consideration of time-scale finer than the

compositing window of 15 days, since few regions are spoiled for cloud-free imagery throughout the entire 15 day window. The adaptive threshold is applied only to the product of the Canny summed edges and the gradient-median-composite images.

**17) Line 148: Is this something you define. If so, make that clear. Otherwise, please provide reference.**

Yes, this is our original algorithm. Will be made clear.

**18) Line 162: In the methods section below Line 162 looks like the start of a discussion to me. I recommend creating a discussion section and placing much of this there. Some of it also belongs more in the introduction perhaps.**

Thanks – we will reconsider the placement of this content.

**19) 185: Like before, no need for parenthesis**

I guess you mean 187 – thanks.

**20) Line 215: Missing space**

Thank you.

**21) Line 244: What indicates this in the plot? The discontinuity in the plot?**

Ah, no – apologies for the confusion. Regions with low automation fraction indicate this. Will be made clear. Reviewer 4 also wondered about the discontinuities. These will now be described in the figure caption.

**22) Line 248: What do you mean with edges vary? The detected edges or the actual ice edge? You mean vary over time when the ice edge is assumed constant? Please explain better.**

Thanks – in hindsight I can see that we can explain this better. We will elaborate to clarify.

**23) Line 263: Please be consistent with the use of notations for in-line lists e.g.**

**1, a, or i.**

Thanks – we will review these for consistency in the revised document.

**24) Line 266: Missing space**

Thank you.

**25) Line 271: Please clarify this sentence as it is not clear what you mean by complexity dataset and linkages between what.**

The dataset we refer to is a dataset of Antarctic coastal margin complexity and configuration, though I agree it doesn't read particularly well as written. Will be clarified.

---

## Author Comment (AC2) · 19 Aug 2020

(Responses separated into general comments to Reviewer 2, "R2A" through "R2C" as a response to the general notes, then "1)", "2)", etc for specific minor comments)

General comments to Reviewer 2: We thank Reviewer 2 for their encouraging, constructive and detailed review. It's particularly heartening to see similar comments to those of the other reviewers, especially Reviewers 1 and 3. We are generally happy to address all of Reviewer 2's comments as detailed below.

**R2: General notes:**

**R2A) I think the spatiotemporal dimensions of the landfast ice datasets you create (1km, 15-day interval) could be better justified. It's entirely reasonable to cite**

the data product and repeat interval of the satellite as reasons for these dimensions. However, because the purpose of this publication is to present a novel dataset for others to use, it would be a good idea to include some discussion of the advantages/disadvantages of these spatiotemporal dimensions.

Good point – we are happy to elaborate a little on this in the text.

As you indicate, our spatial resolution was indeed influenced by the sub-satellite (i.e., nominal) resolution of the thermal infrared channels. Our previous work using fewer swaths per compositing period was limited to a 2 km spatial resolution, but here with more swaths, we were able to get good results with a 1 km spatial resolution.

Regarding the temporal resolution of 15 days, we were driven by a desire to get a finer time-step while still precluding pack ice temporarily advected against the coast from being counted as fast ice. Another factor limiting a finer time-step is cloud coverage. We find that with a 15 day window we are generally able to build high-quality cloud-free composite imagery. We think this is near the limit though – a finer time-step is likely to result in "holes" in the cloud-free composite imagery corresponding to persistently cloudy regions.

**R2B) I understand you are intending to apply this dataset in an analysis for future publication. I would advise you either discuss how these dimensions apply to your intended use of the dataset, or how you envision others using your dataset.**

This would be a good addition to the discussion section. We plan to add a couple of sentences around this.

**R2C) In your results section, for example, you observed an 8.3% increase in fast ice extent compared to Fraser et al.'s (2012) study, which you attributed to the switch from a 20 to 15-day stationary criterion used to identify fast ice. How do these differences in outcomes due to changes in temporal window affect what this data might be used for?**

An extremely good point! We are happy to elaborate on this.

This work has shown that a finer time-step is likely to produce a larger fast ice extent, as expected. This has implications not only for the current work, but also for the next generation of SAR-based observations of fast ice, which, depending on the algorithm, can rely on two observations obtained in subsequent repeat passes. In the case of ESA's SENTINEL-1, this involves a 12-day repeat cycle. TerraSAR-X is shorter still at 11 days, although it has yet to be exploited for fast ice retrieval. Other SAR algorithms which don't rely on exact repeat orbits are able to retrieve fast ice extent over even shorter baselines (e.g., feature-tracking algorithms can deal with any baseline, as long as features are present). These are all likely to retrieve higher fast ice extents than the product here, simply due to the shorter observational baseline.

As you indicate, this probably has implications for end users. This is probably particularly true in regions of ephemeral or volatile fast ice extent. We can mitigate this to some extent with a temporally-continuous dataset such as that presented in this work. For example, we can assess the presence of fast ice across several contiguous time-steps to assess whether a particular region is likely to have volatile fast ice. In such regions, we might suspect that the reported fast ice extent in the region is likely to be higher for a finer time-step.

We plan to add discussion around this to the revised manuscript.

**R2 - Minor comments:**

**1) Line 67: Perhaps provide some examples of these scientific and operational uses?**

A good idea – will be added.

**2) Line 81: The use of parenthesis to clarify the imagery dating back to the year 2000 seems out of place. It would help the flow of the introductory sentence to this section to find a way to integrate this into the sentence without the use of**

**parenthesis.**

Agreed – will be amended.

**3) Line 96: I'm unsure of why "(coastline)" and "(change in)" were inserted into this sentence. Please edit the sentence to make their purpose clear.**

Agreed.

**4) Line 98: The wording of this sentence is a little redundant. You can change it to "Temporal compositing was carried out to create cloud-free. . ." or "Temporal compositing is required to create cloud-free. . .". Since this is the methods section, the reader will already assume this is what you've done, so I would recommend the latter. It is also in keeping with the present tense used in the writing.**

A good point – will be amended.

**5) Line 106: Because the authors are the same for the two studies cited, you can change the in-parenthesis citation to Fraser et al. (2009, 2010).**

I will investigate why my reference manager didn't do this automatically!

**6) Line 107: If you are referring to both cited Fraser et al. studies, say "The earlier works", if you are referring to only one of the cited studies, specify which one.**

Thanks – this is a good way to clarify.

**7) Line 110: Per my comment on line 107, if Fraser et al. 2010 is the work being referenced, make mention of it earlier. Perhaps move this parenthesis citation to the end of the previous sentence that starts with "The earlier work".**

OK – will be clarified.

**8) Line 111: Overall the writing in this manuscript is well done. However there is the tendency to use parenthesis when they are not necessary. The clarification**

that cloud cover is a challenge for optical remote sensing in polar regions can either be integrated into the sentence, or stand as its own sentence. I would recommend the latter, as this would allow for the inclusion of study citations where optical remote sensing was challenging in polar environments.

Thank you for suggesting we revisit our use of parentheses. We will make sure to take a more considered look at them all in the revised manuscript.

**9) Line 118: Please integrate parenthesis into sentence, or create new sentence.**

OK

**10) Line 123: Please integrate parenthesis into sentence.**

Yes, agreed.

**11) Line 126: Please change to "thin clouds" or "thin cloud cover".**

OK – this is in line with Reviewer 1's comment too.

**12) Line 168: Figure 1 is really well done, and does a good job complimenting the written description of the data collection process.**

Thank you! We will also incorporate a flow chart as suggested by Reviewer 3, to further improve comprehension.

**13) Line 182: Parenthesis integrate or remove**

OK

**14) Line 187: Parenthesis integrate or remove**

OK

**15) Lines 192 - 205: I am personally a supporter of numbered lists in publications, especially methods sections, as they are a great help for the audience. However I would recommend some consistency in how these numbered lists are used.**

**From lines 192-205 there are two numbered lists, the first list is independent from the text in a line-by-line format. The second list is integrated in the text. I advise you pick a method of numbering and stick with it. If you choose to integrate both lists into the text, I suggest you separate them into different paragraphs so the readers do not get them confused.**

Good idea. We'll change the format of the second to match the first.

**16) Line 215: Add a space between the final word and the parenthesis containing the citation.**

Thank you, will do.

**17) Line 220-223: This sentence needs work. I would advise either choosing between "ground-breaking" and "new" to avoid redundancy. Remove the parenthesis (across East Antarctica) and integrate into text. Rearrange to improve the flow. For example: "We restrict our presentation of results to the illustration of key attributes in this new pan-Antarctic fast ice dataset, and evaluate its improvements over earlier datasets created for East Antarctica (Fraser et al. 2012).**

Thanks for your help with this paragraph, we agree.

**18) Line 223-224: Remove parenthesis and integrate into text, or delete it. In this case I would advise the latter because the audience already knows you are talking about the dataset you created. Also, the comma separation breaks the flow of the sentence. Try something like: "More in-depth analysis of spatial-temporal patterns and drivers of fast ice distribution is outside the scope of this journal, but is underway for future studies (Fraser et al., in prep.)."**

Thanks – that sounds better.

**19) Line 227: I would remove "important new" adjectives in this sentence when you are referring to the data. The importance has already been demonstrated in the intro and discussion sections, and the purpose of the article is to introduce**

a novel dataset, so the audience already knows it is new.

OK

**20) Line 230: Please integrate the parenthesis text into the sentence.**

OK

**21) Line 233: Please integrate the parenthesis text into the sentence**

OK

**22) Line 236: Please integrate the parenthesis text into the sentence**

OK

**23) Line 259-260: Can you specify any ongoing or anticipated study topics of the Antarctic coastal environment this dataset is expected to help? It's okay if there aren't any that can be specified at the present time, but it would be interesting to include if there are.**

I'm happy to add some examples in the text. These examples include behavioural ecology of charismatic megafauna (e.g., emperor penguin colony presence/absence), the effects of fast ice on the physical oceanography of the continental shelf (e.g., influencing coastal polynya location, and subsequent sea ice production and water mass modification).

**24) Line 260: As I make clear in my summary of this manuscript above, I have no doubt this dataset is a very important contribution to Antarctic fast ice research, and will be heavily cited for years to come. However there is a certain promotional tone in this manuscript that seems out of place in a scientific article. In line 260, "major high-level" is used to emphasize the importance of IPCC reports. However, readers of ESSD will already know the importance and weight of IPCC reports, and will not need these adjectives. Unless "major" and "high-level' are established terms used to organize IPCC reports by importance, I would ad-**
vise leaving them out. Throughout the manuscript you qualify mentions of your data with terms such as "new" and "ground-breaking". While this dataset is indeed new and ground-breaking, it would be better to reserve these terms for sentences when the actual importance of the data is directly addressed, rather than somewhat indiscriminately throughout the manuscript. I understand the purpose of this paper is to make the availability and utility of this new dataset known to the scientific community. I would argue, on your behalf, that the importance of the dataset you created is already evident in your paper, and the scientific community will readily understand this without the need for promotional adjectives.**

Thanks for this perspective. On reflection this language does seem a little out of place here, and would be more suited to a press release, for example. We will rework the text to tone it down.

**25) Lines 263-267: Previously you used numbers when listing steps taken to accomplish a goal. I suggest using numbers here instead of letters, to maintain consistency in the paper.**

OK, we'll adopt this suggestion.

**26) Line 269: In this paper you use the terms "spatial-temporal patterns" and "spatio-temporal patterns". Both are valid terms, but for the sake of consistency I would pick one and use throughout.**

Thanks for picking up on this inconsistency. We'll address it in the revised manuscript.

---

## Author Comment (AC3) · 19 Aug 2020

(Responses separated into general comments to Reviewer 3, "R3A" through "R3D" as a response to the general comments, then "1)", "2)", etc for specific minor comments)

General comments to Reviewer 3: We thank Reviewer 3 for recognising the importance of this dataset, and for their careful and constructive review which will improve the manuscript. We generally agree with all of Reviewer 3's suggestions and happily note that they reflect many of the same suggestions of the other reviewers!

**R3A-D:**

**R3A) I found the description of the algorithm in the methods section somewhat difficult to follow. I would recommend creating a flow-diagram to better illus-**

[Figure]

trate how the algorithm is applied in general This diagram could then refer to Figure 1 to illustrate outputs at various steps in the algorithm. I would also like to see more detail on some aspects of the algorithm. For example, how does the algorithm deal with cases where both thermal and visible imagery are available when generating the 15-day cloud-free composite images? I would also like to see some discussion in the results section on whether there were observed differences between fast ice area products generated from visible and thermal composite images. Further, I would like to see more justification for choosing a 1-km, 15-day epoch for identifying landfast sea ice, and more discussion on how the choice of this epoch influences the generated fast ice extent products.

Similar suggestions were made by Reviewers 1 and 2. The flow chart is a great idea which we will implement.

Regarding your request for more detail, this has also been requested by Reviewers 1 and 2. To specifically answer your questions here:

When visible channel information is available we parallel-process all algorithms for both the Channel 01 (visible) and Channel 31 (thermal IR) cases. Edge guesses are produced for both channels, and combined at the very last step before manual edge completion. We will add this detail to the manuscript. We are also happy to add discussion about the improvements to automation possible when visible channel information is incorporated. The 1 km, 15 day justification has been requested by Reviewer 2 as well – I paste the reply to their comment here for convenience:

As you indicate, our spatial resolution was indeed influenced by the sub-satellite (i.e., nominal) resolution of the thermal infrared channels. Our previous work using fewer swaths per compositing period was limited to a 2 km spatial resolution, but here with more swaths, we were able to get good results with a 1 km spatial resolution.

*Regarding the temporal resolution of 15 days, we were driven by a desire to get a finer time-step while still precluding pack ice temporarily advected against the coast from*

*being counted as fast ice. Another factor limiting a finer time-step is cloud coverage.*
*We find that with a 15 day window we are generally able to build high-quality cloud-free*
*composite imagery. We think this is near the limit though – a finer time-step is likely*
*to result in "holes" in the cloud-free composite imagery corresponding to persistently*
*cloudy regions.*

We didn't perform independent retrieval on visible vs thermal IR input data in times
of both being available. However we note that the performance of the cloud mask is
generally better during times of solar illumination, leading to better quality composite
images, so expect that the automation fraction is generally higher during the summer.

**R3B) I would also like to see more discussion on the fast ice distributions shown**
**in Figure 2. Antarctic fast ice extent can be temporally variable on a regional**
**scale, and I would argue that this variability is not captured by presenting pan-**
**Antarctic maximum and minimum distributions. For example, the fast ice edge**
**in McMurdo Sound in 2016 was significantly farther from the coast than shown**
**in Figure 2 (see, for exampleMYD02.A2016350.0410.006).**

We completely agree with this! However such analysis will appear in our later work,
since it is out of scope for ESSD: "Articles in the data section may pertain to the plan-
ning, instrumentation, and execution of experiments or collection of data. Any interpre-
tation of data is outside the scope of regular articles." (from https://www.earth-system-
science-data.net/about/manuscript$_{t}ypes.html)$

**R3C) The authors state that the number of images contributing to the composite**
**was in- creased relative to the Fraser et al. (2019) algorithm (Lines 114 + 115). I**
**would like to see more details on how this was accomplished, particularly since**
**the epoch was reduced from 20 to 15 days. If I understand correctly, the auto-**
**determined fast ice edge moved an average of âĹij 10 km in a 15-day period. How**
**does this compare to previous regional studies?**

Happy to elaborate on this. Upon clarifying that statement I discovered that our earlier

work (actually Fraser et al., 2010) did indeed use a slightly smaller input image density (number of images per day). However in our earlier work we considered only half as much coast (10 degrees west to 172 degrees east) so the density was in fact probably higher. However in the present work we rank our relatively fewer images more intelligently to ensure more even coverage in all regions (see response to Reviewer 1, relevant response pasted here in italics). We also use the full swath width here, whereas we trimmed in our previous work (which was more susceptible to cloud-mask inaccuracies). Thus we prefer to rewrite point 1 to state "1) ensuring a more even distribution of cloud-free scenes, thereby increasing the chance of a cloud-free view of the surface".

*In line 132 we state that 600 images are incorporated into the composite images for each 15 day period, but we are happy to elaborate on this in the text by saying that these 600 images are separated into 6 regions of 100 images. Without this regional consideration, we found that there is a concentration of images in one or more particular regions based on cloud conditions, since we rank and select the 600 least cloudy granules.*

Yes, we found the auto-determined fast ice edge moves around 10 km in a 15 day period. We aren't aware of any previous regional, automated, long-term datasets but are interested in performing this kind of comparison in future work. Automated SAR products exist but are sporadic in coverage and temporal baseline, so are likely to have a confounded statistic in this regard.

**R3D) The authors state that four adaptive thresholds are set when computing fast ice edge confidence, but then do not describe how these thresholds are utilised in the algorithm. Please provide this detail.**

Apologies for this oversight! These thresholds are used to assign four levels of edge confidence in the automatically-determined edge map. This is the main input to the manual processing step. The manual processing links automatically-determined

edges. This map showing four levels of confidence (as a grey-scale) are particularly helpful in guiding edge completion. This detail will be added to the text at around L150.

**1) Line 7: visible-thermal infrared imagery – change to "compositing visible and thermal infrared imagery".**

OK

**2) Line 38: change ", but at a poorer spatial resolution of 6.25 km (Nihashi and Ohshima, 2015) to limit its" to ", but a poorer spatial resolution of 6.25 km (Nihashi and Ohshima, 2015) limits its"**

Good suggestion, thank you.

**3) Lines 65 – 75: this would fit better in the results section.**

Reviewer 1 also suggested to move it, but to discussion. We will move it to one of these sections.

**4) Line 66: suggest re-order "It also has a multitude of potential scientific and operational uses, given the wide-ranging importance of fast ice" to "Given the wide-ranging importance of fast ice, it also has a multitude of potential scientific and operational uses."**

Good suggestion, we will amend it.

**5) Line 68: remove "developed"**

Agreed.

**6) Line 95: Can you estimate how time intensive it is to update the coastlines and ice shelf edge positions on an annual basis?**

Also a suggestion of Reviewer 2. This update (conducted once per year, or 18 times) was trivial in comparison to edge completion of the 432 fast ice maps. Detail will be added.

**7) Line 96: it is not clear what is meant by "change in".**

Also suggested by Reviewer 2. Will be amended.

**8) Line 104: where are the data provided?**

This is detailed in the abstract, the data availability section and in the reference list, and all three places are mandated by ESSD. I'm hesitant to include the URL again but am happy to if the editor agrees.

**9) Line 139: what is meant by "successive"?**

A mistake by me – also picked up by reviewer 2! This will be amended.

**10) Line 139 + 140: provide more detail by what is meant by "sum over".**

As above – this mistake will be rectified.

**11) Line 142: Provide more detail on how the absolute value of the gradient for the composite image was calculated.**

OK. For each pixel in each composite image (i.e., for the visible composite and the thermal IR composite images separately) the median pixel value was calculated from a 7*7 pixel neighbourhood. Then for each pixel in the median-filtered composite, the magnitude of the gradient vector was obtained. More detail will be added to the text.

**12) Line 149: remove "are set".**

Thank you – will do.

13) Lines 154 – 158: how time intensive is it (on average) to undertake manual processing of fast ice edges? How are the lead-detection images used in the manual processing?

Reviewer 1 also requested this detail. It is the most time-intensive part of the work. One year of manual processing (i.e., 24 maps) can be completed in about one week of approximately full-time work. The aim for the manual processing is to complete

the auto-determined edges, so as to provide a contiguous fast ice edge which can be "bucket-filled" to represent fast ice. This detail will be added to the text.

**14) Line 178: replace "Here and" with "Here, "**

OK

**15) Lines 195 + 197: provide more detail on how the mean fast ice edge separation between composite subsequent images is calculated, e.g. how do you determine which pixel in the second image to "match" with the pixel in the first image?**

We find the nearest edge of similar type (manually- or automatically-determined). Cross-type edges are ignored (i.e., auto to manual, or manual to auto) to avoid confounding results. A cutoff of +/- 50 px (i.e., a 100 km window) is used as an extremely conservative upper bound to avoid the rare case of pixels matching with distant pixels. This detail will be added to the text.

**16) Line 202: explain what is meant by ". . . all remaining manually-determined pixels . . ."**

"Remaining" was a poor choice of word. Sentence changed to "weighting all skeletonised manually-determined pixels by their respective area".

**17) Line 223: replace "journal" with "manuscript".**

We did actually mean "journal" here – ESSD is only for presentation of datasets, not their scientific analysis – but agree that "manuscript" would fit equally well in this case.

**18) Line 248: confirm whether the time period over which these variations have been calculated is 15-days.**

You're right. This clarification will be added.

**Comments on the data set**

[Figure]

**19) In the data set's README file, it states that the latitude of true scale is 70 N. This should read 70 S.**

Thank you for picking up on this error. I have already amended it at the data centre.

---

## Author Comment (AC4) · 19 Aug 2020

General comments to Reviewer 4:

On behalf of the manuscript's authors, I'd like to thank Reviewer 4 for comments to improve our manuscript. In particular, I note that Reviewer 4 has made comments in areas not mentioned by the other reviewers, so I thank you for this contribution.

**1) How have authors distinguished between fast ice and ice shelves from satellites data? with fast ice, it is expected to occasionally collapse at the edge of the ice shelves.**

This process is explained at lines 92 to 99. Particular care was paid to this process, although it is possible that multiyear fast ice was classified as ice shelf in some regions.

[Figure]

We can add this caveat to the text.

**2) Regarding the Figure 4, why are the solid line discontinuous in some places in this figure?**

This is a good question. This discontinuity was described in an early draft of the manuscript but was removed in an effort to keep the manuscript concise! We can spend a few words to describe this in the figure caption. "To remove noise in regions with little fast ice, 1-degree longitude bins with less than 5,000 total fast ice edge pixels across the 18 year dataset were not plotted."

**3) Also, in the discussion section 3.2, the authors have described the results of the comparison between the East and West Antarctic regions. Is it possible to consider the reasons for the differences in terms of sea ice, ocean processes and/or atmospheric fields?**

Absolutely – we strongly agree, and this is very high priority for future work. It's out-of-scope for an ESSD paper, unfortunately (and we think it's good to separate out the science drivers from the dataset anyway): "Articles in the data section may pertain to the planning, instrumentation, and execution of experiments or collection of data. Any interpretation of data is outside the scope of regular articles." (from https://www.earth-system-science-data.net/about/manuscript_types.html)

———————————————————

---

## Author Response (AR1)

**Reviewer 1**

**R1A - Description of methods:**

I am not confident I understand what you have done strictly based on this manuscript. As I think it would be good if this paper stands by itself, I am suggesting that you spend some more time elaborating on the methods.

We have made an extensive overhaul of the datasets and methods section. The "difference" document reveals the extensive changes to this section, but to summarise here:

a) Almost all "methods" suggestions from all reviewers incorporated

b) New flow-chart figure depicting processing pipeline provided as Fig. 1

c) Outcomes of each method step now explicitly stated.

Changes: extensive changes throughout lines 73 to 247. More detail is provided for each specific point below.

- For instance, it is not clear to me how you create the composites. Could you be clearer in terms of this being a mosaic of overlapping imagery or if you are considering just one acquisition for each location and this being a composite of the two channels. Either way, it would be helpful with a couple of sentences addressing how they are created and the number of images typically incorporated.

Addresses, as suggested.

New text at Line 103:

"(i.e., a TIR composite at all times of the year, and a visible composite when solar illumination was present)"

Much more detail now added on the number of images used at Line 112:

"Here we rank all cloud-mask granules by their cloud content, and choose the 100 least cloudy granules in each of six regions (each approximately 60 degrees of longitude wide) around the Antarctic coast for compositing and further processing, i.e., 600 MOD/MYD02 granules in total per 15 day window. This regional consideration was implemented in an effort to ensure a more even distribution of MOD02 granules. We found that without this consideration, the ranking algorithm resulted in a high concentration of granules in a limited number of cloud-free regions at the expense of cloudy regions."

**- The steps in the methods are quite clear in of themselves, but it would help if the outcome of each steps is described as well.**

Addressed, as suggested. Outcomes are given for each step in Lines 139 to 182. The flow chart (new Fig. 1) also aids clarity in this regard.

- This is how I interpret your initial steps: - For any given location and for all 15-day periods in your dataset, you download all available images and create multiple composite images based on available channels (how do you do this?).

More detail provided, as suggested. This detail is included within Lines 100 to 125 (not pasted here, for brevity).

**- Hence, for this location, you have several images (how many roughly?)**

For each 15 day window we use 600 images as an input to all processing. This is now made clearer at Line 112:

"Here we rank all cloud-mask granules by their cloud content, and choose the 100 least cloudy granules in each of six regions (each approximately 60 degrees of longitude wide) around the Antarctic coast for compositing and further processing, i.e., 600 MOD/MYD02 granules in total per 15 day window."

**- Then you detect the edges in all these composites resulting in multiple 1 km resolution (binary datasets?) indicating locations of edges in each composite.**

Addressed – both within the text description, i.e., Line 139 to 182, and the new flow chart (Fig. 1).

**- Then you sum the binary datasets and thus higher numbers more strongly indicate persistent edges at the timescale of 15 days. By doing this you reduce the number of composites down to one product?**

That's right. Well, not quite one product, but we vastly reduce the input images by this process. We retain the Canny vs Sobel edges in separate summary images, for example.

This detail is described within the new flow chart.

**- You then multiply the edges with the median-filtered composite. But which one, if you have multiple composites for this region?**

That's right. This is Step 6 detailed in line 161. Only the (summed) Canny edges are used in this process to construct the edge probability map. This is because the Canny edges have excellent localisation.

**- Also, please spend a sentence on describing how this results in confidence as opposed to just the edge product.**

Addresses, as suggested. Please see Line 167:

"These thresholds are used to construct a grey-scale representation of the edge confidence for each pixel on the grid."

**- What is the range of values prior to normalization?**

Good question. I haven't thought to check, because the normalised product is so much more useful when deploying this algorithm across the whole continent. The lower range is 0 (no Canny edges \* a zero value for the median-filtered composite image). The upper range is typically the product of ~60 Canny edges (a particularly obvious edge with frequent, clear views of the surface throughout the whole period in both Terra and Aqua MODIS) and a composite gradient-median-filtered value of around 5, for the infrared case, or 0.75, in the visible channel case. The IR value is higher due to the numerical value of the brightness temperature difference between cold ice and warm water

being much higher than the difference in reflectance of a dark vs light surface. But these details are helpfully abstracted away thanks to the normalisation.

Since these values are not used in their pre-normalised state, we didn't change the manuscript.

**- Finally, the result is a product of landfast ice edge with 1 km spatial and 15-day temporal resolution. However, how do you eliminate lower confidence edges based on the histogram analysis?**

Edges with a lower confidence than the lowest threshold (which is 98% of the cumulative probability distribution - i.e., the upper 2 percentile of pixels) are eliminated by simple thresholding.

This is made clearer in Line 166-169.

**- If you could attempt to clear up any of misunderstandings and make this a little easier to follow, it would be great. I suggest a figure where the reader can associate each step with a figure panel (or alternately a schematics). Basically, modifying Figure 1 to incorporate the other steps as well**

With these changes, along with the flow diagram suggested by Reviewer 3, we think that a much clearer explanation is now provided.

**R1B:**

**Specific points:**

**-At line 190 you describe: "In the case of a manually-extracted ice edge pixel, it reflects the sum of the ice edge change plus the digitisation error." And you seem to imply that there are no errors in this data?**

I think you meant to type that this implies that there is no error in the automatic digitisation, is that right? This (that the auto-determined pixels have no error) was our initial assumption – since the Canny edge localisation is very good – however you're right - we have implicitly ignored any sub-pixel digitisation error with this assumption.

Performing a quick random point simulation, I can see that the sub-pixel error averages to zero, but has an RMS value of 0.288 px. It could be argued that this value is a better one to use here.

We now use a value of 0.288 px for the automated edge error, and the quadrature sum of this value plus the previously-determined value of 5.47 px (i.e., no 5.48 px) for manually-determined edges. This has now been incorporated into the Methods (Line 223 onward) and Results (Line 300 onward) sections.

**- Could you elaborate about this in the manuscript and discuss the accuracy of the Canny edge detection in general based on your pixel spacing and in terms of misclassification in the case of slow moving non-stationary ice in fjords**

The accuracy of the Canny edge detection is described in the above comment.

Misclassification of melange as fast ice probably occurs in a few limited regions around the coast. We haven't mentioned this in the manuscript because it probably occurs in such limited regions as

to be negligible on a circum-Antarctic scale. But I agree that it's worth noting this caveat in the revised manuscript. Another related error is in regions of densely-packed icebergs, which we pointed out in line 215 of the original manuscript, so this would be a good place to discuss melange misclassification.

This is now included at Line 246: "Regions of ice mélange at the front of ice shelves are another source of uncertainty here, but remain unquantified due to their negligible areal extent on a continental scale."

**- and stationary drifting ice pinned between icebergs or by onshore winds over consecutive 15day periods? I suggest as before to move some of this discussion into either a discussion section or as a subsection to the methods section and discuss caveats a little more deliberately.**

Yes, as you indicate, both issues are probably present to some extent. Regarding the drifting sea ice pinned between grounded icebergs, we have experienced this but only in limited areas in the Antarctic (e.g., visible from stations but at a spatial scale much smaller than one km, our pixel size). Regarding the ice temporarily advected against the shore or existing, genuine fast ice, we still believe that 15 days is long enough to preclude most of this ice from consideration. The coastal flow is generally offshoreward to westward (Turner and Pendlebury, 2001). Blocking anticyclonic pressure systems do occur in southern midlatitudes and these can result in persistent onshoreward winds in particular regions of the Antarctic coast, although the residence time for such systems is rarely longer than one week (Massom et al., 2004).

**New text at Line 126:**

"The 15 day time-step is chosen by balancing a desire for finer resolution against the potential for pack ice temporarily advected against the coast to be misclassified as fast ice despite no mechanical fastening taking place. Around most of coastal Antarctica, the climatological near-surface wind direction is generally offshoreward to westward (Turner and Pendlebury, 2004), thus promoting advection of pack ice away from the coast. Blocking anticyclonic pressure systems do occur in southern mid-latitudes and these can result in persistent onshoreward winds in particular regions of the Antarctic coast, although the residence time for such systems is rarely longer than one week (Massom et al., 2004). As such, a time-step of 15 days is sufficiently long to preclude most of these cases. Drifting sea ice pinned between grounded icebergs may also be misclassified as fast ice, though our earlier work showed that the persistent advection of pack ice into pre-existing coastal features is likely to be a larger problem, and that pack ice held fast between grounded icebergs may quickly become fastened (Fraser et al., 2010)."

J. Turner; S. Pendlebury (Eds.) The International Antarctic Weather Forecasting Handbook, British Antarctic Survey, Cambridge, UK. ISBN: 1855312212, 2001.

R. A. Massom; M. J. Pook; J. C. Comiso; N. Adams; J. Turner; T. Lachlan-Cope; T. T. Gibson. J. Climate (2004) 17 (10): 1914–1928

**- You define landfast ice as stationary for 15 days as opposed to earlier 20 days. A 3-week timeframe is to my knowledge more common. Why did you make this choice and how will this impact the analysis and the potential misclassification of temporarily stationary pack ice etc.**

As above – the circumpolar trough generally permits swift passage of low-pressure systems from west to east. Blocking events can occur north of the ice edge, but these rarely persist more than one week, so 15 days is probably sufficient to exclude this except for extreme cases. Discussion added

to the text at around Line 126, including rationale for the 15 day choice, as in the previous comment.

**- Is the manual delineation very labor intensive e.g. is it sometimes difficult to determine where to draw the line with potentially large consequences for the ice extent? Do you have suggestions for how to mitigate this or how your approach could be improved in the future resulting in a larger than 58\% success rate? If you could discuss this slightly in the manuscript, that would be very interesting.**

The manual delineation ranges from being relatively straightforward (in the case of high quality composite imagery, where few judgement calls need to be made) to quite labour intensive (in the case of heavy cloud obscuring the surface, resulting in ambiguous fast ice edge delineation, and requiring the use of the previous and next 15 day period's composite imagery for guidance). On occasion, such judgement calls have the potential to significantly impact a single period's fast ice extent retrieval in a limited region.

We have taken steps to mitigate this here compared to our earlier work (e.g., by now considering edges visible even under thin cloud; by including more MODIS data per 15-day period). Multisensor fusion would help alleviate this to some extent (we used AMSR-E in our previous work) but limits the time period able to be considered (e.g., AMSR-E was launched 2.5 years after Terra MODIS). Here, our approach is still limited by poor MOD35 cloud mask product accuracy at times. We are interested in implementing state-of-the-art machine-learning cloud masking to mitigate this (e.g., Paul and Huntemann, The Cryosphere Discussions, 2020). This improvement may lead to an automation percentage in excess of the 58\% reported here.

I agree that this kind of discussion is a great addition to the manuscript. We have added text at Lines 281 and 295:

"Manual delineation ranges from being relatively straightforward (in the case of high quality composite imagery, where few judgement-calls need to be made) to quite labour intensive (in the case of heavy cloud obscuring the surface, resulting in ambiguous fast ice edge delineation, and requiring the use of the previous and next 15 day period's composite imagery for guidance). On occasion, such judgement-calls have the potential to significantly impact a single period's fast ice extent retrieval, albeit in a limited region."

**and**

"We have taken steps to mitigate this here compared to our earlier work (e.g., by now considering edges visible even under thin cloud; by more intelligently selecting the least-cloudy MODIS data for each 15 day period). Here, our approach is still limited by relatively poor MOD/MYD35 cloud mask product accuracy at times. In the future, we are interested in implementing state-of-the-art machine-learning cloud masking algorithms to mitigate this (e.g., Paul and Huntemann, 2020). This improvement may lead to an automation percentage in excess of the 58% reported here."

Ref: Paul, S. and Huntemann, M.: Improved machine-learning based open-water/sea-ice/cloud discrimination over wintertime Antarctic sea ice using MODIS thermal-infrared imagery, The Cryosphere Discuss., https://doi.org/10.5194/tc-2020-159, in review, 2020.

**- Could you even provide some speculation into whether other sensors could enhance the analysis?**

Yes – as above, AMSR-E has been used to complement this technique in our previous work (Fraser et al., 2010, which was purely manually-digitised), although it isn't clear how the multisensor fusion could be achieved in the framework of the present paper. Again, we would like to reiterate that few sensors match the very long observational lifetime of MODIS, so a multisensor fusion becomes less attractive in this sense.

**Text added at Line 336:**

"Multi-sensor fusion would help to further mitigate the subjective elements of this dataset to some extent. As an example, we used AMSR-E, in our previous work (Fraser et al., 2010). However, mission overlap generally limits the time period able to be considered in multi-sensor fusion algorithms (e.g., AMSR-E was launched 2.5 years after Terra MODIS, and was effectively decommissioned in 2011)."

**- I realize and appreciate that you have written a quite concise paper and don't want to delve too much into the details. However, I suggest some more clarity and elaboration around these two points.**

We agree that some more detail in these sections would be a useful addition at the cost of a few sentences.

**R1 - Minor comments:**

**1) Line 65 - 74 move to discussion**

Yes, now moved to discussion: Lines 346 to 359.

**2) Line 78: I don't see the "in prep" reference in the bibliography. If not included there, take out.**

Yes, this is still in prep so we will remove it from here.

**3) Table 1: Avoid the word very as in "very high"**

OK – changed in Table 1.

**4) Line 81: Replace "the new" with something like "the fast ice time series presented here" to make it clear that it is not a new one described in Table 1.**

Changed. Now Line 73.

**5) Line 95: Can you clarify how the composites are created? Do you mean creating a mosaic or merging the channels?**

Yes, addressed in the main comment above.

**6) Line 97: Is this manual updating done every year and for the entire coastline? Is this labor intensive if to be done with necessary accuracy?**

Yes, every year for the entire coastline, using the two MOA products (produced in 2004 and 2009) as a baseline. It is relatively quick in comparison to the manual parts of fast ice retrieval. Detail

added to the text in Line 92: "Although this process is entirely manual, it occurs only once per year so is not particularly laborious."

**7) Line 112: "layer of clouds"?**

Amended. Now Line 107.

**8) Line 115: Here as well as prior in the manuscript, the use of parenthesis could be toned down by reformulating.**

Thank you – we note that Reviewer 2 has recommended a parenthesis overhaul too. We have revisited all occurrences in line with these comments.

**9) Line 123: Again, not sure if parenthesis is needed here.**

As above.

**10) Line 126: I am not familiar with the plural form "cloud". Clouds?**

Yes, changed as with comment 7.

**11) Line 125: You have already said this. Take out.**

Thank you - done.

**12) Line 133: Missing oxford comma**

Inserted.

**13) Line 139 and 140: It would be great if you could elaborate here on what you mean by summing edge products. Do you just sum binary pixel values of edge/no edge?**

That's correct. Elaborated on at Lines 151 – 158 by describing outcome:

"– Produce Canny edge images for each granule: Canny edge-detect MOD/MYD02 granules and sum within the current 15 day period. Outcome: Canny edge sum image for automated edge extraction.

– Produce Sobel edge images for each granule (Sobel, 2014): Sobel edge-detect MOD/MYD02 granules and sum within the current 15 day period. Outcome: Sobel edge sum image to guide manual fast ice edge interpretation."

**14) Line 139: Do you mean successive 15-day periods, meaning several periods? If so, how many?**

Sorry, this sentence was explained terribly! Thank you for picking it up. Will be changed from "and sum over successive 15-day periods." to "and summed within the current 15-day period." Same for the following dot point. Line 151.

**15) Line 142: Try to limit redundancy, you have already stated that the composites are cloud-free**

Thank you, redundancy removed.

**16) Line 143: Could you provide a short explanation for the Median filter. For instance, what is this gradient value range? Is there a threshold used to determine whether the edge is stationary and for how long?**

Partly addressed in the response to your main comment above, but to explicitly answer here: The gradient of the median of the composites ranges from 0 to around 5 (for the Channel 31 thermal IR brightness temperature composite) or 0.75 (for the Channel 01 reflectance composite). There is no consideration of time-scale finer than the compositing window of 15 days, since few regions are spoiled for cloud-free imagery throughout the entire 15 day window. The adaptive threshold is applied only to the product of the Canny summed edges and the gradient-median-composite images.

Changes to the main text as described above: Line 155.

**17) Line 148: Is this something you define. If so, make that clear. Otherwise, please provide reference.**

Yes, this is our original algorithm. Made clear in Line 163: "Compute the per-pixel product of the Canny edge image and the gradient-median-composite image described above, which was found to accurately and correctly locate many fast ice edges (i.e., this is an original algorithm)."

**18) Line 162: In the methods section below Line 162 looks like the start of a discussion to me. I recommend creating a discussion section and placing much of this there. Some of it also belongs more in the introduction perhaps.**

This has now been moved to Section 3.2 (the results/discussion)

**19) 185: Like before, no need for parenthesis**

Complete parenthesis overhaul for whole manuscript!

**20) Line 215: Missing space**

Thank you.

**21) Line 244: What indicates this in the plot? The discontinuity in the plot?**

Ah, no – apologies for the confusion. Regions with low automation fraction indicate this. This has now been made clear in Line 291: "By showing longitudes with a low automation fraction, this plot also indicates areas which tend to be most affected by inherent issues detailed in the Methods Section, i.e., persistent cloud cover and/or persistent advection of pack ice toward fast ice that reduces the contrast (in reflectance and surface temperature) between pack and fast ice." Reviewer 4 also wondered about the discontinuities. These are now described in the figure caption.

**22) Line 248: What do you mean with edges vary? The detected edges or the actual ice edge? You mean vary over time when the ice edge is assumed constant? Please explain better.**

This has been clarified at Line 302:

"We find that, on average, manually-determined edges change in location by 5.47 pixels more than that for automatically-determined edges (auto-determined = 10.06 pixels vs manually-determined = 15.53 pixels) in subsequent 15 day windows."

**23) Line 263: Please be consistent with the use of notations for in-line lists e.g. 1, a, or i.**

Thank you – all made consistent now.

**24) Line 266: Missing space**

Thank you.

**25) Line 271: Please clarify this sentence as it is not clear what you mean by complexity dataset and linkages between what.**

The dataset we refer to is a dataset of Antarctic coastal margin complexity and configuration, though I agree it doesn't read particularly well as written. Has been clarified: "Moreover, we plan to study the spatial distribution of fast ice extent in the context of a new dataset describing the multiscale complexity and configuration of the coastline (including aspect) around Antarctica (Porter-Smith et al., in review, 2019), under the hypothesis that the coastal configuration is a first-order determinant of fast ice extent in many regions."

**Reviewer 2: General notes**

R2A) I think the spatiotemporal dimensions of the landfast ice datasets you create (1km, 15day interval) could be better justified. It's entirely reasonable to cite the data product and repeat interval of the satellite as reasons for these dimensions. However, because the purpose of this publication is to present a novel dataset for others to use, it would be a good idea to include some discussion of the advantages/disadvantages of these spatiotemporal dimensions.

As you indicate, our spatial resolution was indeed influenced by the sub-satellite (i.e., nominal) resolution of the thermal infrared channels. Our previous work using fewer swaths per compositing period was limited to a 2 km spatial resolution, but here with more swaths, we were able to get good results with a 1 km spatial resolution.

Regarding the temporal resolution of 15 days, we were driven by a desire to get a finer time-step while still precluding pack ice temporarily advected against the coast from being counted as fast ice. Another factor limiting a finer time-step is cloud coverage. We find that with a 15 day window we are generally able to build high-quality cloud-free composite imagery. We think this is near the limit though – a finer time-step is likely to result in "holes" in the cloud-free composite imagery corresponding to persistently cloudy regions.

New text added:

Line 99: "We choose a 1 km spatial resolution to match the nominal resolution of the MODIS TIR channels."

Line 126: "The 15 day time-step is chosen by balancing a desire for finer resolution against the potential for pack ice temporarily advected against the coast to be misclassified as fast ice despite no mechanical fastening taking place. Around most of coastal Antarctica, the climatological near-surface wind direction is generally offshoreward to westward (Turner and Pendlebury, 2004), thus promoting advection of pack ice away from the coast. Blocking anticyclonic pressure systems do occur in southern mid-latitudes and these can result in persistent onshoreward winds in particular regions of the Antarctic coast, although the residence time for such systems is rarely longer than one week (Massom et al., 2004). As such, a time-step of 15 days is sufficiently long to preclude most of these cases. Drifting sea ice pinned between grounded icebergs may also be misclassified as fast ice, though our earlier work showed that the persistent advection of pack ice into pre-existing coastal features is likely to be a larger problem, and that pack ice held fast between grounded icebergs may quickly become fastened (Fraser et al., 2010). Cloud coverage, which can be persistent in some regions, is a further barrier to a finer time-step when producing visible and TIR composite images of the surface (Fraser et al., 2009)."

**R2B)** I understand you are intending to apply this dataset in an analysis for future publication. I would advise you either discuss how these dimensions apply to your intended use of the dataset, or how you envision others using your dataset.**

This would be a good addition to the summary section. We have added a couple of sentences around this:

**Line 315:**

"Indeed, it is expected to generate and contribute to multiple cross-disciplinary studies of the Antarctic coastal environment. Examples include behavioural ecology of charismatic megafauna (e.g., emperor penguin colony presence/absence), the effects of fast ice on the physical oceanography of the continental shelf (e.g., influencing coastal polynya location, and subsequent sea ice production and water mass modification), and a quantification of the fresh water, nutrients

and biomass within the fast ice itself. Logistical uses are also envisioned (e.g., informing base resupply schedules). Moreover, this dataset directly addresses a key gap identified in major high-level IPCC reports, enabling improved analysis of trends and variability of this key element of the highly-vulnerable Antarctic coastal environment (Vaughan et al., 2013; Meredith et al., 2019)."

**R2C) In your results section, for example, you observed an 8.3% increase in fast ice extent compared to Fraser et al.'s (2012) study, which you attributed to the switch from a 20 to 15-day stationary criterion used to identify fast ice. How do these differences in outcomes due to changes in temporal window affect what this data might be used for?**

An extremely good point! We are happy to elaborate on this.

This work has shown that a finer time-step is likely to produce a larger fast ice extent, as expected. This has implications not only for the current work, but also for the next generation of SAR-based observations of fast ice, which, depending on the algorithm, can rely on two observations obtained in subsequent repeat passes. In the case of ESA's SENTINEL-1, this involves a 12-day repeat cycle. TerraSAR-X is shorter still at 11 days, although it has yet to be exploited for fast ice retrieval. Other SAR algorithms which don't rely on exact repeat orbits are able to retrieve fast ice extent over even shorter baselines (e.g., feature-tracking algorithms can deal with any baseline, as long as features are present). These are all likely to retrieve higher fast ice extents than the product here, simply due to the shorter observational baseline.

As you indicate, this probably has implications for end users. This is probably particularly true in regions of ephemeral or volatile fast ice extent. We can mitigate this to some extent with a temporally-continuous dataset such as that presented in this work. For example, we can assess the presence of fast ice across several contiguous time-steps to assess whether a particular region is likely to have volatile fast ice. In such regions, we might suspect that the reported fast ice extent in the region is likely to be higher for a finer time-step.

**Text added to the manuscript: Line 266:**

"This sensitivity of fast ice extent to observation time-step has implications not only for the current work, but also for the next generation of SAR-based observations of fast ice, which, depending on the algorithm, can rely on two observations obtained in subsequent repeat passes. In the case of ESA's SENTINEL-1, this involves a 12 day repeat cycle. The temporal baseline of DLR's TerraSAR-X is shorter still at 11 days, although it has yet to be exploited for fast ice retrieval. Other SAR-based fast ice retrieval algorithms which don't rely on exact repeat orbits are able to retrieve fast ice extent over even shorter baselines (e.g., feature-tracking algorithms can deal with any baseline, as long as features are present). Such methods are all likely to retrieve higher fast ice extents than the product here, simply due to the shorter observational baseline. As indicated here, differences are particularly strong in regions containing volatile fast ice. As such, end-users of fast ice products in such regions should be cognizant of this phenomenon."

**R2 - Minor comments:**

**1) Line 67: Perhaps provide some examples of these scientific and operational uses?**

**As above – added at Line 315:**

"Indeed, it is expected to generate and contribute to multiple cross-disciplinary studies of the Antarctic coastal environment. Examples include behavioural ecology of charismatic megafauna (e.g., emperor penguin colony presence/absence), the effects of fast ice on the physical oceanography of the continental shelf (e.g., influencing coastal polynya location, and subsequent sea ice production and water mass modification), and a quantification of the fresh water, nutrients and biomass within the fast ice itself. Logistical uses are also envisioned (e.g., informing base resupply schedules)."

2) Line 81: The use of parenthesis to clarify the imagery dating back to the year 2000 seems out of place. It would help the flow of the introductory sentence to this section to find a way to integrate this into the sentence without the use of parenthesis.

Agreed – all bracket use now amended.

3) Line 96: I'm unsure of why "(coastline)" and "(change in)" were inserted into this sentence. Please edit the sentence to make their purpose clear.

Amended along these lines.

4) Line 98: The wording of this sentence is a little redundant. You can change it to "Temporal compositing was carried out to create cloud-free. . ." or "Temporal compositing is required to create cloud-free. . .". Since this is the methods section, the reader will already assume this is what you've done, so I would recommend the latter. It is also in keeping with the present tense used in the writing.

A good point – now amended.

5) Line 106: Because the authors are the same for the two studies cited, you can change the inparenthesis citation to Fraser et al. (2009, 2010).

Now amended – thank you.

6) Line 107: If you are referring to both cited Fraser et al. studies, say "The earlier works", if you are referring to only one of the cited studies, specify which one.

Thanks – this is a good way to clarify.

7) Line 110: Per my comment on line 107, if Fraser et al. 2010 is the work being referenced, make mention of it earlier. Perhaps move this parenthesis citation to the end of the previous sentence that starts with "The earlier work".

OK – clarified.

8) Line 111: Overall the writing in this manuscript is well done. However there is the tendency to use parenthesis when they are not necessary. The clarification that cloud cover is a challenge for optical remote sensing in polar regions can either be integrated into the sentence, or stand as its own sentence. I would recommend the latter, as this would allow for the inclusion of study citations where optical remote sensing was challenging in polar environments.

Thank you for suggesting we revisit our use of parentheses. We have taken a more considered look at them all in the revised manuscript.

**9) Line 118: Please integrate parenthesis into sentence, or create new sentence.**

**10) Line 123: Please integrate parenthesis into sentence.**

Yes, amended.

**11) Line 126: Please change to "thin clouds" or "thin cloud cover".**

Done – this is in line with Reviewer 1's comment too.

**12) Line 168: Figure 1 is really well done, and does a good job complimenting the written description of the data collection process.**

Thank you! We have also incorporated a flow chart as suggested by Reviewer 3, to further improve comprehension.

**13) Line 182: Parenthesis integrate or remove**

OK

14) Line 187: Parenthesis integrate or remove

OK

15) Lines 192 - 205: I am personally a supporter of numbered lists in publications, especially methods sections, as they are a great help for the audience. However I would recommend some consistency in how these numbered lists are used. From lines 192-205 there are two numbered lists, the first list is independent from the text in a line-by-line format. The second list is integrated in the text. I advise you pick a method of numbering and stick with it. If you choose to integrate both lists into the text, I suggest you separate them into different paragraphs so the readers do not get them confused.

Good idea. We have changed the format of the second to match the first.

16) Line 215: Add a space between the final word and the parenthesis containing the citation.

Thank you, done.

17) Line 220-223: This sentence needs work. I would advise either choosing between "groundbreaking" and "new" to avoid redundancy. Remove the parenthesis (across East Antarctica) and integrate into text. Rearrange to improve the flow. For example: "We restrict our presentation of results to the illustration of key attributes in this new pan-Antarctic fast ice dataset, and evaluate its improvements over earlier datasets created for East Antarctica (Fraser et al. 2012).

Thanks for your help with this paragraph, we agree. This has been amended.

18) Line 223-224: Remove parenthesis and integrate into text, or delete it. In this case I would advise the latter because the audience already knows you are talking about the dataset you created. Also, the comma separation breaks the flow of the sentence. Try something like: "More in-depth analysis of spatial-temporal patterns and drivers of fast ice distribution is outside the scope of this journal, but is underway for future studies (Fraser et al., in prep.)."

Thanks – that sounds better.

**19)** Line 227: I would remove "important new" adjectives in this sentence when you are referring to the data. The importance has already been demonstrated in the intro and discussion sections, and the purpose of the article is to introduce a novel dataset, so the audience already knows it is new.

OK

20) Line 230: Please integrate the parenthesis text into the sentence.

OK

21) Line 233: Please integrate the parenthesis text into the sentence

OK

22) Line 236: Please integrate the parenthesis text into the sentence

OK

**23) Line 259-260: Can you specify any ongoing or anticipated study topics of the Antarctic coastal environment this dataset is expected to help? It's okay if there aren't any that can be specified at the present time, but it would be interesting to include if there are.**

I'm happy to add some examples in the text: Line 317

"Examples include behavioural ecology of charismatic megafauna (e.g., emperor penguin colony presence/absence), the effects of fast ice on the physical oceanography of the continental shelf (e.g., influencing coastal polynya location, and subsequent sea ice production and water mass modification), and a quantification of the fresh water, nutrients and biomass within the fast ice itself. Logistical uses are also envisioned (e.g., informing base resupply schedules)."

24) Line 260: As I make clear in my summary of this manuscript above, I have no doubt this dataset is a very important contribution to Antarctic fast ice research, and will be heavily cited for years to come. However there is a certain promotional tone in this manuscript that seems out of place in a scientific article. In line 260, "major high-level" is used to emphasize the importance of IPCC reports. However, readers of ESSD will already know the importance and weight of IPCC reports, and will not need these adjectives. Unless "major" and "high-level" are established terms used to organize IPCC reports by importance, I would advise leaving them out. Throughout the manuscript you qualify mentions of your data with terms such as "new" and "ground-breaking". While this dataset is indeed new and ground-breaking, it would be better to reserve these terms for sentences when the actual importance of the data is directly addressed, rather than somewhat indiscriminately throughout the manuscript. I understand the purpose of this paper is to make the availability and utility of this new dataset known to the scientific community. I would argue, on your behalf, that the importance of the dataset you created is already evident in your paper, and the scientific community will readily understand this without the need for promotional adjectives.

Thanks for this perspective. On reflection this language does seem a little out of place here, and would be more suited to a press release, for example. We have reworked the text throughout the manuscript to tone it down.

**25) Lines 263-267: Previously you used numbers when listing steps taken to accomplish a goal. I suggest using numbers here instead of letters, to maintain consistency in the paper.**

OK, we have adopted this suggestion.

26) Line 269: In this paper you use the terms "spatial-temporal patterns" and "spatio-temporal patterns". Both are valid terms, but for the sake of consistency I would pick one and use throughout.

Thanks for picking up on this inconsistency. We have chosen to use "spatio-temporal" throughout.

**Reviewer 3**

Main comments R3A-D:

R3A) I found the description of the algorithm in the methods section somewhat difficult to follow. I would recommend creating a flow-diagram to better illustrate how the algorithm is applied in general This diagram could then refer to Figure 1 to illustrate outputs at various steps in the algorithm. I would also like to see more detail on some aspects of the algorithm. For example, how does the algorithm deal with cases where both thermal and visible imagery are available when generating the 15-day cloud-free composite images? I would also like to see some discussion in the results section on whether there were observed differences between fast ice area products generated from visible and thermal composite images. Further, I would like to see more justification for choosing a 1-km, 15-day epoch for identifying landfast sea ice, and more discussion on how the choice of this epoch influences the generated fast ice extent products.

Similar suggestions were made by Reviewers 1 and 2. The flow chart is a great idea which we have implemented (Fig 1).

Regarding your request for more detail, this has also been requested by Reviewers 1 and 2. To specifically answer your questions here:

When visible channel information is available we parallel-process all algorithms for both the Channel 01 (visible) and Channel 31 (thermal IR) cases. Edge guesses are produced for both channels, and combined at the very last step before manual edge completion. We have added this detail to the manuscript. We are also happy to add discussion about the improvements to automation possible when visible channel information is incorporated. The 1 km, 15 day justification has been requested by Reviewer 2 as well – I paste the reply to their comment here for convenience:

As you indicate, our spatial resolution was indeed influenced by the sub-satellite (i.e., nominal) resolution of the thermal infrared channels. Our previous work using fewer swaths per compositing period was limited to a 2 km spatial resolution, but here with more swaths, we were able to get good results with a 1 km spatial resolution.

Regarding the temporal resolution of 15 days, we were driven by a desire to get a finer time-step while still precluding pack ice temporarily advected against the coast from being counted as fast ice. Another factor limiting a finer time-step is cloud coverage. We find that with a 15 day window we are generally able to build high-quality cloud-free composite imagery. We think this is near the limit though – a finer time-step is likely to result in "holes" in the cloud-free composite imagery corresponding to persistently cloudy regions.

We didn't perform independent retrieval on visible vs thermal IR input data in times of both being available. However we note that the performance of the cloud mask is generally better during times of solar illumination, leading to better quality composite images, so expect that the automation fraction is generally higher during the summer.

Changes made:

Figure 1 is new flow chart.

Complete overhaul of methods section: Line 73 to 247 – too numerous to list - see "diff" document for complete list.

R3B) I would also like to see more discussion on the fast ice distributions shown in Figure 2. Antarctic fast ice extent can be temporally variable on a regional scale, and I would argue that this variability is not captured by presenting pan-Antarctic maximum and minimum distributions. For example, the fast ice edge in McMurdo Sound in 2016 was significantly farther from the coast than shown in Figure 2 (see, for exampleMYD02.A2016350.0410.006).

We completely agree with this! However such analysis will appear in our later work, since it is out of scope for ESSD: "Articles in the data section may pertain to the planning, instrumentation, and execution of experiments or collection of data. Any interpretation of data is outside the scope of regular articles." (from <a href="https://www.earth-system-science-data.net/about/manuscript\_types.html">https://www.earth-system-science-data.net/about/manuscript\_types.html</a>)

**R3C) The authors state that the number of images contributing to the composite was increased relative to the Fraser et al. (2019) algorithm (Lines 114 + 115). I would like to see more details on how this was accomplished, particularly since the epoch was reduced from 20 to 15 days. If I understand correctly, the auto-determined fast ice edge moved an average of $\sim$ 10 km in a 15-day period. How does this compare to previous regional studies?**

Happy to elaborate on this. Upon clarifying that statement I discovered that our earlier work (actually Fraser et al., 2010) did indeed use a slightly smaller input image density (number of images per day). However in our earlier work we considered only half as much coast (10 degrees west to 172 degrees east) so the density was in fact probably higher. However in the present work we rank our relatively fewer images more intelligently to ensure more even coverage in all regions (see response to Reviewer 1, relevant response pasted here in italics). We also use the full swath width here, whereas we trimmed in our previous work (which was more susceptible to cloud-mask inaccuracies). Thus we prefer to rewrite point 1 to state "1) ensuring a more even distribution of cloud-free scenes, thereby increasing the chance of a cloud-free view of the surface".

In line 132 we state that 600 images are incorporated into the composite images for each 15 day period, but we are happy to elaborate on this in the text by saying that these 600 images are separated into 6 regions of 100 images. Without this regional consideration, we found that there is a concentration of images in one or more particular regions based on cloud conditions, since we rank and select the 600 least cloudy granules.

**New text added to manuscript: (Line 112)**

"Here we rank all cloud-mask granules by their cloud content, and choose the 100 least cloudy granules in each of six regions (each approximately 60 degrees of longitude wide) around the Antarctic coast for compositing and further processing, i.e., 600 MOD/MYD02 granules in total per 15 day window. This regional consideration was implemented in an effort to ensure a more even distribution of MOD02 granules. We found that without this consideration, the ranking algorithm resulted in a high concentration of granules in a limited number of cloud-free regions at the expense of cloudy regions."

Yes, we found the auto-determined fast ice edge moves around 10 km in a 15 day period. We aren't aware of any previous regional, automated, long-term datasets but are interested in performing this kind of comparison in future work. Automated SAR products exist but are sporadic in coverage and temporal baseline, so are likely to have a confounded statistic in this regard.

R3D) The authors state that four adaptive thresholds are set when computing fast ice edge confidence, but then do not describe how these thresholds are utilised in the algorithm. Please provide this detail.

Apologies for this oversight! These thresholds are used to assign four levels of edge confidence in the automatically-determined edge map. This is the main input to the manual processing step. The manual processing links automatically-determined edges. This map showing four levels of confidence (as a grey-scale) are particularly helpful in guiding edge completion. This dot point has been changed to:

"Produce a normalised histogram of edge confidence, setting four adaptive thresholds at 0.995 (highest-confidence edge), 0.990, 0.985 and 0.980 (lowest-confidence edge). These thresholds are used to construct a grey-scale representation of the edge confidence for each pixel on the grid. Outcome: Confidence-classified automated fast ice edge map."

**1) Line 7: visible-thermal infrared imagery – change to "compositing visible and thermal infrared imagery".**

OK

**2) Line 38: change ", but at a poorer spatial resolution of 6.25 km (Nihashi and Ohshima, 2015) to limit its" to ", but a poorer spatial resolution of 6.25 km (Nihashi and Ohshima, 2015) limits its"**

Good suggestion, thank you.

**3) Lines 65 – 75: this would fit better in the results section.**

Reviewer 1 also suggested this. We moved it to the results/brief discussion section.

4) Line 66: suggest re-order "It also has a multitude of potential scientific and operational uses, given the wide-ranging importance of fast ice" to "Given the wide-ranging importance of fast ice, it also has a multitude of potential scientific and operational uses."

Good suggestion, we have amended it.

**5) Line 68: remove "developed"**

Agreed.

**6) Line 95: Can you estimate how time intensive it is to update the coastlines and ice shelf edge positions on an annual basis?**

Also a suggestion of Reviewer 2. This update (conducted once per year, or 18 times) was small in comparison to edge completion of the 432 fast ice maps. Detail has been added: Line 92: "Although this process is entirely manual, it occurs only once per year so is not particularly laborious."

**7) Line 96: it is not clear what is meant by "change in".**

Also suggested by Reviewer 2. Has been amended.

**8) Line 104: where are the data provided?**

This is detailed in the abstract, the data availability section and in the reference list, and all three places are mandated by ESSD. We decided not to add this detail around Line 104 because of this.

Instead, we moved the sentence "All data are provided as Climate and Forecast (CF)-compliant NetCDF files." to the "data availability" section (Line 273).

**9) Line 139: what is meant by "successive"?**

A mistake by me – also picked up by reviewer 2! This has been amended.

**10) Line 139 + 140: provide more detail by what is meant by "sum over".**

As above – this mistake has now been rectified.

**11) Line 142: Provide more detail on how the absolute value of the gradient for the composite image was calculated.**

OK. For each pixel in each composite image (i.e., for the visible composite and the thermal IR composite images separately) the median pixel value was calculated from a 7\*7 pixel neighbourhood. Then for each pixel in the median-filtered composite, the magnitude of the gradient vector was obtained. This is now explicitly stated at Line 142: "Produce gradient-median-composite images: Median-filter (using a 7\*7 pixel sliding window) each composite image (i.e., visible and TIR), then take the absolute value of the gradient of this image, indicating edges in the composite image. Outcome: Gradient-mean-composite images for automated edge extraction."

**12) Line 149: remove "are set".**

Thank you – removed.

**13) Lines 154 – 158: how time intensive is it (on average) to undertake manual processing of fast ice edges? How are the lead-detection images used in the manual processing?**

Reviewer 1 also requested this detail. It is the most time-intensive part of the work. One year of manual processing (i.e., 24 maps) can be completed in about one week of approximately full-time work. The aim for the manual processing is to complete the auto-determined edges, so as to provide a contiguous fast ice edge which can be "bucket-filled" to represent fast ice. This detail has been added to the text, Line 172: "Carry out necessary manual processing (relatively labour-intensive, one year takes approximately 40 hours):"

**14) Line 178: replace "Here and" with "Here, "**

OK

**15) Lines 195 + 197: provide more detail on how the mean fast ice edge separation between composite subsequent images is calculated, e.g. how do you determine which pixel in the second image to "match" with the pixel in the first image?**

We find the nearest edge of similar type (manually- or automatically-determined). Cross-type edges are ignored (i.e., auto to manual, or manual to auto) to avoid confounding results. A cutoff of +/- 50 px (i.e., a 100 km window) is used as an extremely conservative upper bound to avoid the rare case of pixels matching with distant pixels. This detail has been added to the text. Line 214: "We find the nearest edge of similar type. In this step, we match automatically-determined edge pixels with the nearest automatically-determined edge in the subsequent image. Cross-type edge matches are ignored (i.e., auto to manual, or manual to auto) to avoid confounding results. A cutoff of +/- 50 px

(i.e., an  $\sim$ 100 km window) is used as an extremely conservative upper bound to avoid the rare case of pixels matching with distant pixels."

**16) Line 202: explain what is meant by "... all remaining manually-determined pixels ..."**

"Remaining" was a poor choice of word. Sentence changed to "weighting all skeletonised manually-determined pixels by their respective area". (Line 229)

**17) Line 223: replace "journal" with "manuscript".**

We did actually mean "journal" here – ESSD is only for presentation of datasets, not their scientific analysis – but agree that "manuscript" would fit equally well in this case. Changed to "manuscript".

**18) Line 248: confirm whether the time period over which these variations have been calculated is 15-days.**

You're right. This clarification has been added.

**Comments on the data set**

**19)** In the data set's README file, it states that the latitude of true scale is 70 N. This should read 70 S.**

Thank you for picking up on this error. I have amended it at the data centre.

**Reviewer 4**

**1) How have authors distinguished between fast ice and ice shelves from satellites data? with fast ice, it is expected to occasionally collapse at the edge of the ice shelves.**

This process is explained at lines 84 to 94. Particular care was paid to this process, although it is possible that multiyear fast ice was classified as ice shelf in some regions. This caveat has been added to the text: "It is possible that some very persistent multi-year fast ice is misclassified as ice shelf in limited regions, although particular care was paid to avoid this." (Line 93)

**2) Regarding the Figure 4, why are the solid line discontinuous in some places in this figure?**

This is a good question. This discontinuity was described in an early draft of the manuscript but was removed in an effort to keep the manuscript concise! We can spend a few words to describe this in the figure caption. "To remove noise in regions with little fast ice, 1-degree longitude bins with less than 5,000 total fast ice edge pixels across the 18 year dataset were not plotted."

**3) Also, in the discussion section 3.2, the authors have described the results of the comparison between the East and West Antarctic regions. Is it possible to consider the reasons for the differences in terms of sea ice, ocean processes and/or atmospheric fields?**

Absolutely – we strongly agree, and this is very high priority for future work. It's out-of-scope for an ESSD paper, unfortunately (and we think it's good to separate out the science drivers from the dataset anyway): "Articles in the data section may pertain to the planning, instrumentation, and execution of experiments or collection of data. Any interpretation of data is outside the scope of regular articles." (from <a href="https://www.earth-system-science-data.net/about/manuscript">https://www.earth-system-science-data.net/about/manuscript</a> types.html)

**High-resolution mapping of circum-Antarctic landfast sea ice distribution, 2000-2018**

Alexander D. Fraser1,2,3, Robert A. Massom4,2, Kay I. Ohshima3, Sascha Willmes5, Peter J. Kappes6, Jessica Cartwright7,8,2, and Rick Porter-Smith2

[revised manuscript text omitted]